# DIFFERENTIAL PRIVACY OF HYBRID QUANTUM-CLASSICAL ALGORITHMS

## ABSTRACT

Differential privacy has been successfully used to safeguard the privacy of classical algorithms and has more recently been extended to protect the privacy of quantum algorithms. However, in the present era of Noisy Intermediate-Scale Quantum (NISQ) computing, practical applications are limited to hybrid quantum-classical algorithms (e.g., quantum machine learning and variational quantum algorithms) to tackle computational tasks due to inherent quantum noise. Unfortunately, the issue of privacy in such algorithms has been largely disregarded. This paper addresses this gap by defining the differential privacy of quantum measurements as a means to protect the overall privacy of hybrid quantum-classical algorithms. The core concept involves the use of differentially private quantum measurements to ensure privacy since hybrid quantum-classical algorithms heavily rely on quantum measurements for the interaction between quantum and classical computing. To address this, we explore post-processing and composition theorems to establish the efficiency and feasibility of differentially private quantum measurements. By introducing quantum depolarizing noise or a unique classical noise (measurement-based exponential mechanisms) into quantum measurements, we bolster the security of algorithms against privacy violations. Taking the hybrid nature of differentially private quantum measurements, our framework offers both classical and quantum differential privacy. To validate these theoretical results, we carry out various numerical experiments demonstrating the effectiveness and practicality of our framework using differentially private quantum measurements to protect the privacy of hybrid quantum-classical algorithms.

## 1 INTRODUCTION

In the present era of *Noisy Intermediate-Scale Quantum (NISQ)* computing Preskill (2018), characterized by quantum computers containing hundreds of noisy quantum bits (qubits), noise inevitably impacts the quantum computing process. Consequently, there has been a surge in the development of *hybrid quantum-classical algorithms*. These algorithms leverage the complementary strengths of classical and quantum computing to address the limited computational capacity resulting from medium scalability and noise in NISQ computers. Hybrid quantum-classical algorithms operate akin to classical machine learning algorithms, utilizing parameterized quantum circuits (analogous to classical neural networks) alongside classical optimizers to adjust parameters based on the output distributions of quantum data obtained from quantum measurements, thereby enabling the resolution of intricate computational tasks. Notable examples of such algorithms include the Variational Quantum Eigensolver (VQE) Peruzzo et al. (2014) and the Quantum Approximate Optimization Algorithm (QAOA) Farhi et al. (2014). VQE is tailored for determining the ground state energy of molecules and materials. It accomplishes this by generating potential states using parameterized quantum circuits and subsequently employing classical optimizers to minimize the expected value. QAOA, on the other hand, concentrates on addressing combinatorial optimization problems by producing feasible solutions through parameterized quantum circuits and utilizing classical optimizers to refine parameters to optimize the objective function. Consequently, hybrid quantum-classical algorithms present a promising approach for integrating quantum computing into practical applications within the current NISQ era. They offer advantages in terms of performance and resilience to noise Jones et al. (2019); Endo et al. (2021).

In algorithms like these, the handling of privacy-sensitive classical data (e.g., personal financial records and drug information) stored in quantum data is crucial, as highlighted in several studies Cao et al. (2018); De Leon et al. (2021); Orús et al. (2019). This heightened awareness emphasizes the importance of protecting users' privacy within these algorithms. In classical algorithms, addressing personal privacy concerns often involves employing differential privacy Dwork et al. (2014), which aims to reduce the impact of individual data differences in neighboring datasets on algorithm outcomes. The concept of differential privacy has also been utilized to improve quantum data privacy in quantum information processing by establishing meaningful relationships between neighboring quantum states. These relationships are mainly assessed using informative metrics – local operation Aaronson & Rothblum (2019) and trace distance Zhou & Ying (2017) – to describe the similarity between quantum states qualitatively and quantitatively, respectively. Local operation involves a classical similarity achieved through a local operation to transition one state to another, while trace distance measures the difference between quantum states on a scale from 0 to 1. As a result, there is a growing body of literature exploring quantum differential privacy Senekane et al. (2017); Watkins et al. (2023); Hirche et al. (2023); Angrisani et al. (2022); Angrisani & Kashefi (2022); Quek et al. (2021); Du et al. (2021); Nuradha et al. (2024) in various scenarios. However, these studies predominantly focus on the quantum realm and often overlook privacy concerns in hybrid quantum-classical algorithms where quantum and classical information are exchanged through quantum measurements.

In this paper, we focus on ensuring the privacy guarantee of hybrid quantum-classical algorithms by designing differentially private quantum measurements used in such algorithms. Our approach addresses the limitations of existing classical and quantum differential privacy frameworks, which primarily target privacy leakage in either purely classical or purely quantum data processing (See Appendix A for details). To achieve this, we study how to enforce differential privacy for quantum measurements by introducing appropriate privacy-enhancing mechanisms. To ensure that privacy guarantees are preserved even after further processing, we establish a post-processing theorem. While quantum measurements inherently involve randomness, they often do not satisfy strong differential privacy conditions. To strengthen privacy protection, we introduce classical noise mechanisms applied after measurements and quantum noise mechanisms applied before measurements. Specifically, we propose using either quantum depolarizing noise (analogous to the classical randomized response mechanism Mironov (2017)) or a novel measurement-based exponential mechanism (an adaptation of the classical exponential mechanism Dwork et al. (2014)). The measurement-based exponential mechanism uses the original measurement outcome distributions as utility functions, allowing outcome selection with differential privacy guarantees without relying on sensitivity calculations. Furthermore, we establish a composition theorem to analyze the privacy loss when multiple quantum measurements are combined. Finally, through extensive numerical experiments, we demonstrate the effectiveness and practicality of our proposed methods for protecting the differential privacy of quantum measurements in hybrid quantum-classical algorithms.

In summary, our main contributions are as follows:

1. *Formulating* a method to protect the differential privacy of quantum measurements in hybrid quantum-classical algorithms, thereby ensuring both classical and quantum differential privacy guarantees.

2. *Introducing* a post-processing theorem to guarantee the robustness of differentially private quantum measurements against subsequent computations following the initial analysis of private data.

3. *Utilizing* either a quantum depolarizing noise or (classical) measurement-based exponential mechanism to enhance the privacy of quantum measurements to a desired privacy protection budget.

4. *Developing* a composition theorem to establish the differential privacy of hybrid quantum-classical algorithms by utilizing jointly differentially private quantum measurements.

5. *Performing* a series of numerical experiments to validate the efficiency of our framework, particularly focusing on the efficacy of integrating differentially private quantum measurements through quantum depolarizing noise and measurement-based exponential mechanisms.

## 2 Preliminaries

In this section, we review the basic concepts and notations of hybrid quantum-classical algorithms and quantum/classical differential privacy. For more details, we refer to Nielsen & Chuang (2001).

Let $\mathcal{E}$ denote a quantum (noisy) circuit or channel (i.e., a completely positive trace-preserving map) acting on quantum states of a Hilbert space $\mathcal{H}$, and let $\mathcal{M} = \{\Pi_i\}_{i \in \mathcal{O}}$ represent a quantum measurement, where $\{\Pi_i\}$ is a POVM (positive operator-valued measure) on $\mathcal{H}$ with outcome set $\mathcal{O}$.

**Definition 1** *A hybrid quantum-classical algorithm $\mathcal{A} = (\mathcal{E}, \mathcal{M} = \{\Pi_i\}_{i \in \mathcal{O}})$ on a Hilbert space $\mathcal{H}$ is a randomized function $\mathcal{A} : \mathcal{D}(\mathcal{H}) \to \mathcal{O}$ from quantum state set $\mathcal{D}(\mathcal{H})$ to classical measurement outcome set $\mathcal{O}$ satisfying the measurement outcome distribution:*
$$p_i = \Pr[\mathcal{A}(\rho) = i] = \mathrm{tr}(\Pi_i \mathcal{E}(\rho)) \quad \forall i \in \mathcal{O}, \rho \in \mathcal{D}(\mathcal{H}).$$

An illustration of a hybrid quantum-classical algorithm is shown in Figure 1.

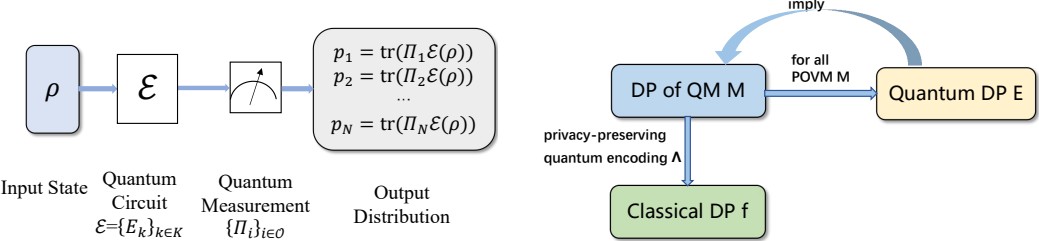

Figure 1: Hybrid quantum-classical algorithm.  Figure 2: Relationship among DP frameworks.

**Heisenberg Picture:** To analyze quantum measurements in hybrid quantum-classical algorithms, we adopt the Heisenberg picture, which contrasts with the Schrödinger picture by focusing on the evolution of observables rather than quantum states. Given a quantum measurement $\mathcal{M} = \{\Pi_i\}_{i \in \mathcal{O}}$ and a quantum circuit $\mathcal{E}$ with Kraus operators $\{E_j\}$, the measurement outcome probability can be equivalently expressed as:
$$\Pr[\mathcal{M}(\mathcal{E}(\rho)) = i] = \mathrm{tr}(\mathcal{E}^\dagger(\Pi_i)\rho),$$
where $\mathcal{E}^\dagger(\cdot) = \sum_j E_j^\dagger(\cdot)E_j$ is the adjoint of $\mathcal{E}$. This reformulation defines a new measurement $\mathcal{M}_\mathcal{E} = \{\mathcal{E}^\dagger(\Pi_i)\}_{i \in \mathcal{O}}$, allowing the action of a quantum circuit followed by measurement to be interpreted as measuring $\rho$ directly with $\mathcal{M}_\mathcal{E}$. This perspective is particularly useful for analyzing privacy-preserving mechanisms on quantum measurements.

**Classical and Quantum Differential Privacy.** Differential privacy (DP) was originally used to protect individual privacy in classical datasets and has been extended to quantum settings. In both frameworks, privacy is defined concerning neighboring inputs, but the notion of "neighboring" differs.

**Definition 2 (Classical Differential Privacy Dwork et al. (2014))** *Let $\epsilon \geq 0$ and $0 \leq \delta < 1$. A randomized function $\mathcal{K}$ satisfies $(\epsilon, \delta)$-classical differential privacy (CDP) if for all neighboring datasets $\vec{\nu} \sim \vec{\omega}$ and any subset $S \subseteq Range(\mathcal{K})$,*
$$\Pr[\mathcal{K}(\vec{\nu}) \in S] \leq e^\epsilon \Pr[\mathcal{K}(\vec{\omega}) \in S] + \delta,$$
*where $\vec{\nu} \sim \vec{\omega}$ means the two datasets differ in only one entry.*

**Definition 3 (Quantum Differential Privacy Zhou & Ying (2017))** *Let $\epsilon \geq 0$ and $0 \leq \delta < 1$. A quantum channel $\mathcal{E}$ satisfies $(\epsilon, \delta)$-quantum differential privacy (QDP) if for all neighboring quantum states $\rho \sim \sigma$, any measurement $\mathcal{M} = \{\Pi_i\}_{i \in \mathcal{O}}$, and any subset $S \subseteq \mathcal{O}$,*
$$\sum_{i \in S} \mathrm{tr}(\Pi_i \mathcal{E}(\rho)) \leq e^\epsilon \sum_{i \in S} \mathrm{tr}(\Pi_i \mathcal{E}(\sigma)) + \delta.$$

Several notions of neighboring quantum states have been proposed. The two most widely used are: (1) **Trace distance-based Zhou & Ying (2017):** $\rho \sim \sigma$ if $\frac{1}{2}\mathrm{tr}|\rho - \sigma| \leq \eta$, for some fixed $\eta \in [0, 1]$. (2) **Local operation-based Aaronson & Rothblum (2019):** $\rho \sim \sigma$ if they can transfer to each other by a 1-qubit operation. A hybrid formulation combining both definitions was proposed in Angrisani et al. (2023).

## 3 DIFFERENTIAL PRIVACY OF HYBRID QUANTUM-CLASSICAL ALGORITHM

In this section, we study the differential privacy of quantum measurements within hybrid quantum-classical algorithms. Since quantum measurements serve as the interface between quantum and classical computation, their privacy properties directly impact the overall privacy of the algorithm. To rigorously characterize and protect the privacy of such measurements, we develop post-processing and composition theorems, enabling practical analysis and modular design of privacy-preserving mechanisms. A central focus of our approach is the construction of differentially private quantum measurements, which are the core component for achieving privacy of hybrid quantum-classical algorithms. For convenience, all proofs of our theoretical results are put in the appendix E.

Now, we begin with defining differential privacy for quantum measurements.

**Definition 4 (Differential Privacy of Quantum Measurement)** *Let $\epsilon \geq 0$ and $1 > \delta \geq 0$ be constants. A quantum measurement $\mathcal{M} = \{\Pi_i\}_{i \in \mathcal{O}}$ is said to satisfy $(\epsilon, \delta)$-differential privacy ($(\epsilon, \delta)$-DP) if for any pair of neighboring quantum states $\rho \sim \sigma$ and any subset $S \subseteq \mathcal{O}$, the following inequality holds:*

$$\Pr[\mathcal{M}(\rho) \in S] \leq \exp(\epsilon) \cdot \Pr[\mathcal{M}(\sigma) \in S] + \delta.$$

*Here, $\Pr[\mathcal{M}(\rho) \in S] = \sum_{i \in S} \Pr[\mathcal{M}(\rho) = i] = \sum_{i \in S} \operatorname{tr}(\Pi_i \rho)$ represents the probability that the measurement outcome belongs to a subset $S$ of $\mathcal{O}$. When $\delta = 0$, the quantum measurement satisfies the standard $\epsilon$-differential privacy condition.*

This definition ensures that the probability distributions over outcomes resulting from measuring two neighboring quantum states are similar up to a multiplicative factor $\exp(\epsilon)$ and additive term $\delta$. It is conceptually aligned with classical and quantum differential privacy, while specifically focusing on the privacy of fixed quantum measurements in hybrid settings.

We leave the notion of neighboring quantum states abstract at this point. Any reasonable definition can be applied in this context and in subsequent results, such as the post-processing and composition theorems. If a specific definition is required, it will be clearly stated in the relevant sections.

The definition of differential privacy for quantum measurements integrates key ideas from CDP and QDP, as introduced in Definitions 2 and 3, respectively. This connection is illustrated in Figure 2, which demonstrates how the measurement-based definition serves as a conceptual bridge between the classical and quantum settings with the help of the neighboring-preserving quantum encoding method, which encodes classical information into quantum states keeping neighboring relationship between classical and quantum settings and is formally introduced in the appendix B. We provide a precise characterization of this property in the following theorem.

**Theorem 1** *Let $\mathcal{E}$ represent a quantum circuit and let $\Lambda$ be a neighboring-preserving quantum encoding. It follows that:*

- *$\mathcal{E}$ is $(\epsilon, \delta)$-QDP if and only if for any quantum measurement $\mathcal{M} = \{\Pi_i\}_{i \in \mathcal{O}}$, the transformed measurement $\mathcal{M}_{\mathcal{E}} = \{\mathcal{E}^\dagger(\Pi_i)\}_{i \in \mathcal{O}}$ in the Heisenberg picture is $(\epsilon, \delta)$-DP.*

- *If a quantum measurement $\mathcal{M}$ is $(\epsilon, \delta)$-DP, then the classical randomized function $\mathcal{M} \circ \Lambda$ is $(\epsilon, \delta)$-CDP.*

*Here, $\mathcal{M} \circ \Lambda$ denotes the functional composition of $\mathcal{M}$ and $\Lambda$.*

Table 1: Comparison of CDP, QDP, and differential privacy of quantum measurement.

| Privacy Type | Input | Differentially Private Mechanism | Output |
|---|---|---|---|
| CDP | Classical Dataset | Randomized Function | Classical Distribution |
| QDP | Quantum State[*] | Quantum Circuit | Quantum State |
| DP of QM | Quantum State[*] | Quantum Measurement | Classical Distribution |

[*] The quantum state can also be derived from classical data through quantum encoding.

Based on the above theorem, the definition of differential privacy for quantum measurements enables connections to both classical and quantum differential privacy. Specifically, classical differential

privacy can be recovered by applying a neighboring-preserving quantum encoding, while quantum differential privacy can be achieved by introducing quantum noise that ensures differential privacy under any quantum measurement. To avoid confusion among these three definitions, Table 1 presents a comparison highlighting their key differences.

Recall that incorporating a randomized mechanism can protect the privacy of classical algorithms. However, while quantum measurement inherently involves randomness due to the probabilistic aspect of quantum computing, its primary purpose is not to ensure privacy protection. Hence, quantum measurement typically does not offer an effective guarantee of differential privacy. To illustrate this point more clearly, we provide a representative example in Appendix C, which highlights the need for developing differentially private quantum measurements to effectively safeguard the privacy of hybrid quantum-classical algorithms. Given the mixed nature of quantum measurements, incorporating both quantum and classical noise is essential. Similar to classical and quantum differential privacy frameworks, it is crucial to establish post-processing and composition theorems initially to ensure the efficiency and scalability of differentially private quantum measurements in the following section.

### 3.1 POST-PROCESSING THEOREM AND COMPOSITION THEOREM

Post-processing is a crucial aspect of differential privacy as it prevents any additional computational analysis by an adversary from divulging more information about an individual's privacy. Just like in classical computing, the preservation of differential privacy of quantum measurements relies on the post-processing theorem.

**Theorem 2** *Let $\mathcal{M} = \{\Pi_i\}_{i \in \mathcal{O}}$ be an $(\epsilon, \delta)$-DP quantum measurement on a Hilbert space $\mathcal{H}$. For any randomized function $\mathcal{K} : \mathcal{O} \to \mathcal{O}'$, $\mathcal{K} \circ \mathcal{M} : \mathcal{D}(\mathcal{H}) \to \mathcal{O}'$ is $(\epsilon, \delta)$-DP.*

This theorem demonstrates that classical post-processing does not affect overall privacy guarantees. This is critical, as it assures that adversarial manipulations do not compromise privacy.

In classical computing, the composition of multiple algorithms plays a fundamental role in building more complex systems. Correspondingly, composition theorems have been established for both classical and quantum differential privacy Dwork et al. (2014); Guan et al. (2023) to ensure that privacy guarantees are preserved under composition. As part of our framework, we introduce a composition theorem that formally characterizes the privacy guarantees when combining differentially private quantum measurements.

**Theorem 3 (informal)** *If quantum measurements $\mathcal{M}_1 = \{\Pi_i\}_{i \in \mathcal{O}_1}$ and $\mathcal{M}_2 = \{\Pi_j\}_{j \in \mathcal{O}_2}$ are $(\epsilon_1, \delta_1)$-DP and $(\epsilon_2, \delta_2)$-DP respectively, then the combined measurement is $(\epsilon_1 + \epsilon_2, \delta_1 + \delta_2)$-DP.*

The formal format of the above theorem is provided in Appendix D.

## 4 DIFFERENTIALLY PRIVATE QUANTUM MEASUREMENTS

In this section, we describe how to design differentially private quantum measurements by introducing classical or quantum noise. The overall framework is illustrated in Fig. 3.

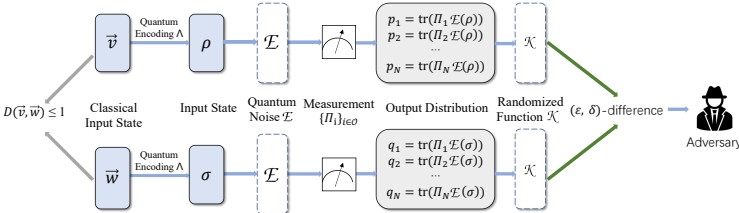

Figure 3: Our hybrid quantum-classical differential privacy framework

**Classical Noise Strategy.** In CDP, Laplace and Gaussian noise are effective for continuous, real-valued data. However, quantum measurement outcomes are discrete and finite, limiting the applicability of these methods. Prior work has explored their use in quantum settings Angrisani et al. (2023; 2022), their applicability is limited due to this mismatch in suitability.

Therefore, we propose the *measurement-based exponential mechanism (MBEM)* as a classical noise approach well-suited to quantum measurements. This method addresses the discrete nature of measurement outcomes and aims to balance privacy and utility. MBEM adapts the classical exponential mechanism Dwork et al. (2014) to the quantum setting, enabling differentially private quantum measurements. The mechanism selects outputs based on utility scores, assigning probabilities via an exponential function, which is particularly effective for categorical outcomes like measurement results. Further details are provided in Appendix F. In our adaptation, the original measurement outcome distribution is used as the utility function, and outcomes are redistributed accordingly. This extends the exponential mechanism to the quantum domain, preserving its privacy-utility guarantees in probabilistic settings.

Below, we outline the step-by-step details of our MBEM.

1. *Defining measurement-based utility function:* Given a quantum measurement $\mathcal{M} = \{\Pi_i\}_{i \in \mathcal{O}}$, the outcome distribution for input state $\rho$ is $P_{\mathcal{M}}(i|\rho) = \text{tr}(\Pi_i \rho)$. This probability serves as the utility function: $u(\rho, i) = P_{\mathcal{M}}(i|\rho)$. For large-scale computations where the exact output distribution is difficult to evaluate, one can instead employ the empirical distribution obtained from repeated measurements as the utility function.

2. *Setting sensitivity bound:* Sensitivity $\Delta u$ represents the maximum impact of the utility function $u$ concerning changes in the neighboring quantum states and outcome set $\mathcal{O}$. As $u$ is a distribution, we can set a global maximum $\Delta u = 1$, independent of the neighboring relation between quantum states.

3. *Calculating probabilities:* Following the exponential mechanism, the probability of selecting outcome $i$ is

$$P_{EM}(i|\rho) = \frac{\exp\left(\frac{\epsilon u(\rho, i)}{2 \Delta u}\right)}{\sum_{j \in \mathcal{O}} \exp\left(\frac{\epsilon u(\rho, j)}{2 \Delta u}\right)} \tag{1}$$

4. *Selecting measurement outcome:* By using the calculated probability distribution $P_{EM}(i|\rho)$, a measurement outcome is randomly chosen from the set $\mathcal{O}$.

The procedures outlined above guarantee that our MBEM can offer $\epsilon$-DP.

**Theorem 4** *The MBEM defined through the above procedure can enable any quantum measurement to be $\epsilon$-DP.*

**Quantum Noise Strategy.** We turn to introduce quantum noise to enable differentially private quantum measurements. We opt for quantum depolarizing noise as the method for achieving this, given its ability to operate independently of the target quantum measurement and the quantum state neighboring relationships. By utilizing this noise, we can enhance the assurance of achieving better differential parivacy.

Let's delve into the concept of quantum noise. A quantum noise, denoted as $\mathcal{E}_N$, consists of a set of Kraus matrices $\{E_{N,k}\}_{k \in K}$ and operates as $\mathcal{E}_N(\rho) = \sum_{k \in K} E_{N,k} \rho E_{N,k}^\dagger$ for all $\rho \in \mathcal{D}(\mathcal{H})$.

For an $n$-qubit Hilbert space $\mathcal{H}$, the depolarizing noise $\mathcal{E}_{\text{Dep}}$ can be expressed as:

$$\mathcal{E}_{\text{Dep}}(\rho) = (1-p)\rho + \frac{pI}{\text{Dim}},$$

where $\text{Dim} = 2^n$ represents the dimension of the state Hilbert space, and $p$ is the noisy probability.

To assess this enhancement quantitatively, we will utilize trace distance-based neighboring relationships between quantum states in the following discussion of this section, that is, $\rho \sim \sigma$ if and only if the trace distance $\tau(\rho, \sigma) = \frac{1}{2}\text{tr}|\rho - \sigma| \leq \eta$.

Building upon this concept, we first prove that introducing any quantum noise $\mathcal{E}_N$ will not degrade the differential privacy of any quantum measurement $\mathcal{M}$.

**Theorem 5** *If a quantum measurement $\mathcal{M} = \{\Pi_i\}_{i \in \mathcal{O}}$ is $(\epsilon, \delta)$-DP, then $\mathcal{M}_{\mathcal{E}_N} = \{\mathcal{E}_N^\dagger(\Pi_i)\}_{i \in \mathcal{O}}$ is also $(\epsilon, \delta)$-DP for any quantum noise $\mathcal{E}_N$.*

This theorem demonstrates that incorporating quantum noise does not compromise privacy protection, even if the quantum noise is not generated by a differential privacy mechanism (a designated noise), but rather arises from the intrinsic random noise in quantum devices or a combination thereof. This substantiates the efficacy of our framework in the existing NISQ era where random noise is an inevitable factor of hybrid quantum-classical algorithms.

Considering Theorem 5 enables the exploration of introducing noise, prompting the inquiry into the possibility of utilizing noise to achieve differential privacy in quantum measurements. The affirmative answer to this lies in the utilization of depolarizing noise. This is formally stated in the following.

**Theorem 6** *Suppose we have a quantum measurement denoted by $\mathcal{M} = \{\Pi_i\}_{i \in \mathcal{O}}$ and a depolarizing noise denoted by $\mathcal{E}_{Dep}$ with a noisy probability $p$. Then the obtained noisy measurement $\mathcal{M}_{\mathcal{E}_{Dep}}$ is $(\epsilon, \bar{\delta})$-DP for any $\epsilon \geq 0$, where*

$$\bar{\delta} = \max_{S \subseteq \mathcal{O}} \left[ (1-p)\gamma_S - (e^\epsilon - 1)\frac{p\,\mathrm{tr}(\Pi_S)}{Dim} \right]$$

*with $\gamma_S = \eta\lambda_{\max}(\Pi_S) - (e^\epsilon + \eta - 1)\lambda_{\min}(\Pi_S)$ and $\Pi_S = \sum_{k \in S} \Pi_k$. If we focus on $\epsilon$-DP, then $\mathcal{M}_{\mathcal{E}_{Dep}}$ is $\bar{\epsilon}$-DP, where $\bar{\epsilon} = \ln[(\bar{\theta} - 1)\eta + 1]$ and $\bar{\theta} = \max_{S \subseteq \mathcal{O}} \theta_S$ with*

$$\theta_S = \frac{Dim \cdot (1-p)\lambda_{\max}(\Pi_S) + \mathrm{tr}(\Pi_S)p}{Dim \cdot (1-p)\lambda_{\min}(\Pi_S) + \mathrm{tr}(\Pi_S)p}. \tag{2}$$

By the above result, we can adjust the noisy probability $p$ to achieve the intended privacy budget $\bar{\epsilon}$ and $\bar{\delta}$ in the differentially private noisy measurement $\mathcal{M}_{\mathcal{E}_{Dep}}$.

The result of Theorem 6 relies on the quantum measurement $\mathcal{M}$. If the details of the measurement are unknown or if we aim to ensure that all quantum measurements are differentially private, we can still employ depolarizing noise to accomplish this with a worse (higher) $\epsilon$-DP.

**Corollary 1** *Let $\mathcal{E}_{Dep}$ represent a depolarizing noise characterized by a noisy probability $p$. For any quantum measurement $\mathcal{M}$, the obtained noisy measurement $\mathcal{M}_{\mathcal{E}_{Dep}}$ is $\epsilon$-DP, where $\epsilon = \ln\left(\frac{Dim \cdot (1-p)}{p}\eta + 1\right)$.*

In Corollary 1, we present a general upper bound on the differential privacy guarantee achievable for any quantum measurement. This result aligns with prior findings in the QDP framework when depolarizing noise is applied to quantum circuits Zhou & Ying (2017), indicating that our measurement-level privacy mechanism can also satisfy QDP-style guarantees.

Furthermore, our approach remains effective under alternative notions of neighboring quantum states, such as the local operation-based definition. Since the maximum trace distance between quantum states is 1, setting $\eta = 1$ in Theorem 6 ensures that all quantum states are treated as neighbors. Under this setting, the depolarizing noise channel $\mathcal{E}_{Dep}$ guarantees a privacy level of $\ln\left(\frac{Dim \cdot (1-p)}{p} + 1\right)$ for any measurement, regardless of the underlying notion of proximity between states.

**Remark.** In the above, we propose a measurement-based method to ensure the differential privacy of a hybrid quantum-classical algorithm by introducing noise through both quantum and classical noise mechanisms. Theorems 4 and 6 demonstrate that either the Measurement-Based Exponential Mechanism (MBEM) or depolarizing noise can significantly improve privacy guarantees. We validate our approach using a series of quantum machine learning (QML) algorithm examples based on variational quantum classifiers (VQC), from Examples 2 to 4 in the appendix C, demonstrating how each mechanism transforms the privacy characteristics of the measurement.

## 5  EVALUATION

In this section, to evaluate the effectiveness and implications of these mechanisms, we conduct a series of numerical experiments. These experiments assess both the privacy-utility trade-offs of MBEM and depolarizing noise across different parameter settings and provide internal comparisons between the mechanisms. In our experiments, we quantify utility loss using the maximum Kullback-Leibler (KL) divergence between measurement outcome distributions before and after noise addition; see Appendix H for the details. Additionally, we compare our approach against the existing QDP framework, highlighting that achieving equivalent privacy guarantees often requires less noise when using our proposed methods for specified quantum measurements.

## 5.1 HYBRID DIFFERENTIAL PRIVACY MECHANISMS

We evaluate the efficacy of the MBEM and the depolarizing noise mechanism in achieving $\epsilon$-DP across various $\epsilon$ values.

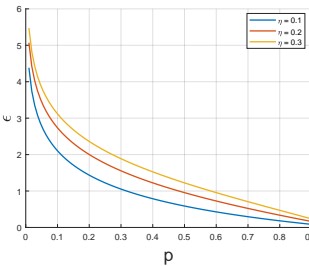
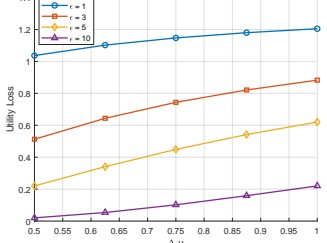
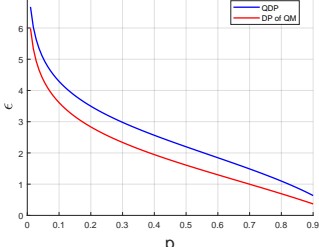

Figure 4: Trade-off between $\epsilon$-DP and $p$ of depolarizing noise for neighboring value $\eta$.

Figure 5: Trade-off between utility loss and MBEM sensitivity $\Delta u$ for different $\epsilon$-DP.

Figure 6: Comprison of $\epsilon$-QDP and $\epsilon$-DP in terms of noisy probability p.

*Quantum depolarizing noise mechanism.* To assess how depolarizing noise influences privacy, we examine the relationship between the noise probability $p$ and privacy parameter $\epsilon$, as per Corollary 1, with results shown in Fig. 4. Experiments are conducted on a 3-qubit system (Dim $= 8$), using trace distance-based $\eta$-neighboring relationships for various $\eta$ values. Results show that increasing $p$ consistently reduces $\epsilon$, indicating stronger privacy, regardless of the $\eta$ value.

We also investigate the relationship between $\eta$ and $\epsilon$, as presented in Appendix J.2, which highlights the trade-off between neighborhood looseness and privacy budget.

*Measurement-based exponential mechanism.* To evaluate MBEM, we analyze the relationship between utility loss and the sensitivity parameter $\Delta u$, as shown in Fig. 5. The figure presents four curves under different privacy budgets $\epsilon$, illustrating how utility loss varies with $\Delta u$. Since $\Delta u$ measures the sensitivity of probability distributions, it is bounded above by 1, with a tighter upper bound of $\Delta u = 1/2$ in our setting. Thus, Fig. 5 spans $\Delta u \in [1/2, 1]$. The results show that as $\Delta u$ decreases (i.e., becomes more precise), utility loss also decreases. This indicates that more accurate choices of $\Delta u$ can improve utility while maintaining privacy, confirming MBEM's effectiveness in balancing the trade-off.

Additional experiments and analysis on $\Delta u$ and its relationship with $\epsilon$ are provided in Appendix J.1, further highlighting how optimizing $\Delta u$ can yield better privacy-utility trade-offs.

## 5.2 COMPARISON TO QUANTUM DIFFERENTIAL PRIVACY

Our approach differs from QDP in scope: QDP ensures privacy under all possible quantum measurements, while we focus on a fixed, known measurement (see Fig. 2). Furthermore, when applying quantum depolarizing noise, our method offers stronger privacy protection than the QDP framework. This difference is visualized in Fig. 6, which compares the privacy parameter estimated by QDP with the actual differential privacy parameter achieved under our method, as discussed in Example 4. The figure shows a clear gap between the two, highlighting that our method achieves tighter privacy bounds when the measurement is fixed and known. This demonstrates a significant advantage of our approach in the context of hybrid quantum-classical algorithms.

## 5.3 PRIVACY-UTILITY TRADE-OFF ANALYSIS

We conduct an empirical investigation into the trade-off between privacy and utility by comparing our MBEM and depolarizing noise mechanisms internally as well as with other mechanisms across various quantum circuits featuring different numbers of qubits. In each comparison, we present the most representative experiment, while additional experimental findings are detailed in Appendix . These experiments collectively highlight the benefits of our mechanisms in effectively balancing privacy and utility.

We performed experiments utilizing three different kinds of quantum circuits: *GHZ circuit*, *variational quantum circuit (VQC)* and *random circuit (RC)*. VQCs represent a common hybrid quantum-classical algorithm extensively applied in quantum machine learning. They function as quantum counterparts

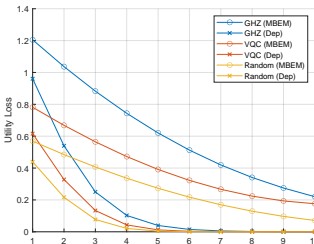

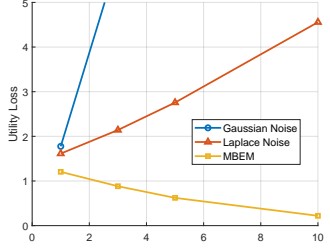

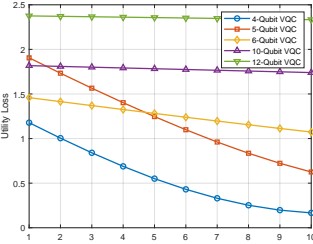

Figure 7: Privacy-utility trade-off for MBEM and depolarizing noise mechanisms (Dep) on three 3-qubit circuits.

Figure 8: Comparison of the privacy-utility trade-off of different classical noise mechanisms on a GHZ circuit.

Figure 9: Privacy-utility curves for MBEM on variational quantum circuits (VQC) with varied qubit numbers.

to classical neural networks, facilitating parameter tuning for particular objectives. The random circuit is generated in a random manner based on a predefined pattern.

**Internal Comparison:** We initially assess the balance between privacy and utility in our MBEM utilizing classical noise and the depolarizing noise mechanism employing quantum noise. The trade-offs between privacy and utility for both approaches on three 3-qubit quantum circuits are illustrated in Fig. 7. The graph clearly illustrates that both MBEM and the depolarizing noise mechanism result in a reduction in utility as the level of privacy protection increases. However, the extent of this reduction varies among the different circuits. Additionally, we notice that the depolarizing noise mechanism, incorporating quantum noise, exhibits better utility compared to the MBEM method using classical noise, at the same level of privacy protection. Nonetheless, the correlation between utility and privacy for these methods does not exhibit a significant distinction. This suggests that both methods are valid choices for privacy protection, and the selection between them should be based on the specific requirements of the practical issue at hand.

**MBEM v.s. Other Classical Noise:** We evaluate the effectiveness of our MBEM approach in comparison to the commonly employed Laplace mechanism and the Gaussian noise mechanism within our framework. The specific implementation details can be found in the Appendix I. By plotting privacy-utility curves, illustrated in Fig. 8, we examine the trade-offs between privacy and utility. The results indicate that the MBEM method offers notably improved utility without compromising the level of privacy provided. Additionally, it is apparent that the utility of MBEM increases as the level of privacy protection lessens.

**Qubit Number:** At last, the privacy-utility trade-offs for MBEM and depolarizing noise mechanisms on quantum random circuits and variational quantum circuits (VQC) with varied qubit numbers are conducted. The result of MBEM on VQC is presented in Fig. 9 , and other similar results are presented in Appendix K. In these experiments, VQC and RC allow for a range of 3 to 12 qubits. Although our approach is applicable to any qubit number, the computation of utility loss becomes more resource-intensive with increasing qubit numbers due to the exponential growth in the dimension of the measurement outcome distribution. To streamline operations, we present the privacy-utility trade-off analysis within the confines of 12 qubits. The results highlight that our methods can be applied to any quantum circuits with different qubit number.

## 6 CONCLUSION

In this paper, we have introduced a quantum measurement-based differential privacy framework to address privacy concerns in hybrid quantum-classical algorithms. The post-processing theorem we have presented ensures the maintenance of privacy guarantees from quantum measurements even after subsequent data processing. Additionally, our composition theorem establishes a comprehensive method for ensuring the differential privacy of complex hybrid quantum-classical algorithms. By leveraging the unique characteristics of quantum measurements, our approach integrates quantum depolarizing noise with the measurement-based exponential mechanism to provide robust privacy protections, as demonstrated through a series of numerical experiments. Moreover, we have demonstrated how our framework enables achieving both classical and quantum differential privacy. These contributions extend the application scope of differential privacy techniques into the realm of hybrid quantum-classical algorithms, bridging a significant gap in the current literature.

## ETHICS STATEMENT

Consistent with the ICLR Code of Ethics, we confirm that this work does not raise any ethical concerns. The research is purely theoretical and does not involve human subjects, sensitive data, or applications with potential ethical, legal, or societal implications.

## REPRODUCIBILITY STATEMENT

We have made every effort to ensure the reproducibility of our results. All theoretical analyses are provided in detail in the main text and appendix. Furthermore, the complete source code and all data required to reproduce the reported experiments will be made available in the supplementary materials.

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

## A  RELATED WORKS

Classical and quantum differential privacy methods have effectively protected the privacy of classical and quantum algorithms, respectively. Nevertheless, these approaches may not directly ensure the privacy of present hybrid quantum-classical algorithms due to their limitations outlined below. In this work, we address the differential privacy of quantum measurements in hybrid quantum-classical algorithms and propose effective mechanisms to tackle these challenges.

**Classical Methods:** *Classical differential privacy (CDP)* Dwork et al. (2014) has undergone extensive research across various fields such as data analysis, machine learning, and dataset queries Abadi et al. (2016); Murakami & Kawamoto (2019); Hu et al. (2021); Gadotti et al. (2022); Avent et al. (2017); Xiao & Xiong (2015); Li et al. (2017). These classical methods ensure robust privacy safeguards of deterministic algorithms by introducing classical randomized noise mechanisms to their outputs. Key noise mechanisms include the Laplace mechanism, Gaussian mechanism, and exponential mechanism, each offering distinct strategies for achieving CDP. The selection between these mechanisms depends on the specific data characteristics and privacy objectives. The Laplace/Gaussian mechanism involves adding noise sampled from the Laplace/Gaussian distribution to the output of a deterministic function to enforce differential privacy. However, these mechanisms are unsuitable for developing differentially private quantum measurements due to the inherent randomness in the measurements. Instead, we consider adapting the exponential mechanism into the quantum domain for implementing differentially private quantum measurements as it works for categorical outputs that can be regarded as measurement outcomes. The exponential mechanism upholds differential privacy by selecting outputs based on their quality scores, with probabilities determined by an exponential function of the score. To apply this mechanism to quantum measurements, we leverage the original measurement outcome probability distribution as quality scores for measurement outcomes, redistributing outcomes based on the exponential function of the scores. This extension broadens the utility of the exponential mechanism to the quantum realm within the probabilistic domain. Through the innovative measurement-based exponential mechanism, the creation of differentially private quantum measurements is simplified, reinforcing the privacy of hybrid quantum-classical algorithms.

**Quantum Methods:** Researchers have been exploring how to incorporate differential privacy into quantum algorithms Hirche et al. (2023); Angrisani et al. (2022); Angrisani & Kashefi (2022); Quek et al. (2021); Du et al. (2021). They are investigating the impact of inherent quantum noise in quantum algorithms and additional quantum noise on the overall differential privacy of these algorithms. Specifically, *quantum differential privacy (QDP)* is designed to protect the output states of quantum algorithms against privacy breaches resulting from any quantum measurement. In our context of a hybrid quantum-classical algorithm, where only one specific measurement is revealed, the focus is on mitigating privacy risks associated with the particular measurement rather than all potential quantum measurements. To tackle this issue, a depolarizing mechanism is suggested to introduce quantum noise directly into the quantum measurement. This approach simplifies the process of achieving differential privacy for the measurement, ensuring a desired level of privacy protection for the hybrid quantum-classical algorithm. This level of protection depending on the measurement offers a higher degree (lower bound) of privacy compared to traditional QDP, as demonstrated in Theorem 6 and its corollary.

## B  NEIGHBORING-PRESERVING QUANTUM ENCODING

As discussed in Section 2, various quantum encoding techniques exist for transforming classical datasets into quantum states. When a quantum encoding method can maintain the neighboring relationships present in the original classical datasets within the resulting quantum states, it is referred to as a *neighboring-preserving quantum encoding* technique Angrisani et al. (2023); Guan et al. (2023). In other words, if two classical datasets $\vec{\nu}, \vec{\omega}$ are encoded using a neighboring-preserving quantum encoding $\Lambda$ to produce quantum states $\Lambda(\vec{\nu})$ and $\Lambda(\vec{\omega})$, the neighboring relationship between the original classical datasets should be preserved in the quantum domain, implying that

$$\vec{\nu}, \vec{\omega} \text{ are neighboring} \Rightarrow \Lambda(\vec{\nu}), \Lambda(\vec{\omega}) \text{ are neighboring}.$$

An illustration showcasing the implementation of a neighboring-preserving quantum encoding through amplitude encoding, can be found in the Appendix B.1.

In addition to utilizing amplitude encoding for implementing a neighboring-preserving quantum encoding approach, basis encoding can also be employed for classical bit string datasets, maintaining local operation-based neighboring relationships introduced in Section 2. Basis encoding inherently converts classical bit strings to quantum bit strings; for instance, "0001" is represented as $|0001\rangle \langle 0001|$ through basis encoding. The local operation-based neighboring relationship is essentially an extension of its classical counterpart.

However, trying to create a neighboring-preserving quantum encoding through basis encoding and a trace distance-based neighboring relationship would be unproductive and inconsequential. This is because, in basis encoding, setting $\eta = 1$ is required to achieve this, as the trace distance between quantum states encoding any two neighboring bit strings is 1. Since the trace distance ranges from 0 to 1, all quantum states are considered neighboring states, making the concept less significant. Therefore, it is essential to precisely define neighboring relationships of quantum states, especially when exploring various quantum encoding techniques, particularly in the context of utilizing quantum computing for addressing classical problems with differential privacy guarantees.

Now, by leveraging neighboring-preserving quantum encoding and the Heisenberg picture as outlined in Section 2, we can connect CDP and QDP through our framework, as demonstrated in the following theorem and depicted in Fig. 2.

## B.1 Neighboring-preserving example

**Example 1** *Consider two neighboring classical vector datasets $\vec{\nu} = (\nu_0, ..., \nu_{n-1}), \vec{\omega} = (\omega_0, ..., \omega_{n-1}) \in \Omega$, differing only in a single element, let's say the $(k+1)$-th element, i.e. $\nu_k \neq \omega_k$. Initially, we normalize these vectors to obtain $\frac{\vec{\nu}}{\|\vec{\nu}\|}$ and $\frac{\vec{\omega}}{\|\vec{\omega}\|}$, where $\|\vec{\nu}\|$ and $\|\vec{\omega}\|$ represent the norms of $\vec{\nu}$ and $\vec{\omega}$, respectively. Subsequently, employing the amplitude encoding, these vectors are transformed into quantum states $\Lambda(\frac{\vec{\nu}}{\|\vec{\nu}\|})$ and $\Lambda(\frac{\vec{\omega}}{\|\vec{\omega}\|})$, correspondingly.*

*The trace distance between these states is given by:*

$$\tau\left(\Lambda(\frac{\vec{\nu}}{\|\vec{\nu}\|}), \Lambda(\frac{\vec{\omega}}{\|\vec{\omega}\|})\right) = \sqrt{1 - \left|\frac{\vec{\nu}^\dagger}{\|\vec{\nu}\|} \cdot \frac{\vec{\omega}}{\|\vec{\omega}\|}\right|^2}.$$

*Let $M = \max_{\vec{\nu} \in \Omega} \max_i \frac{|\nu_i|^2}{\|\vec{\nu}\|^2}$ denote the maximum square norm of an element among all normalized vectors. Then assuming $|\nu_k| \geq |\omega_k|$, we obtain:*

$$\begin{aligned}
\left|\frac{\vec{\nu}^\dagger}{\|\vec{\nu}\|} \cdot \frac{\vec{\omega}}{\|\vec{\omega}\|}\right|^2 &= \left|\frac{\nu_0\omega_0 + ... + \nu_{n-1}\omega_{n-1}}{\|\vec{\nu}\|\|\vec{\omega}\|}\right|^2 \\
&= \left|\frac{\|\vec{\nu}\|^2 - |\nu_k|^2 + \nu_k\omega_k}{\|\vec{\nu}\|\|\vec{\omega}\|}\right|^2 \\
&\geq \left|\frac{\|\vec{\nu}\|^2 - 2|\nu_k|^2}{\|\vec{\nu}\|^2}\right|^2 \\
&\geq (1 - 2M)^2.
\end{aligned}$$

*Consequently, we derive:*

$$\tau(\Lambda(\frac{\vec{\nu}}{\|\vec{\nu}\|}), \Lambda(\frac{\vec{\omega}}{\|\vec{\omega}\|})) \leq \sqrt{4M - 4M^2}.$$

*Given the arbitrariness of $\vec{\nu}$ and $\vec{\omega}$, we can set $\eta = \sqrt{4M - 4M^2}$ to maintain the classical neighboring relationship in the trace distance-based $\eta$-neighboring relationship within the encoded quantum states through amplitude encoding. Thus, we have successfully implemented a neighboring-preserving quantum encoding utilizing amplitude encoding.*

# C  RUNNING EXAMPLES

## C.1  DIFFERENTIAL PRIVACY FOR VQC

*Variational Quantum Classifier (VQC).* Variational quantum circuits (VQCs) have become a central component in quantum machine learning, particularly for classification tasks. A VQC is a parameterized quantum circuit designed to process classical data encoded into quantum states and produce measurement outcomes that can be used to distinguish between data classes. It typically consists of an input encoding layer (e.g., angle or amplitude encoding), followed by trainable quantum gates and entangling layers, and ends with measurement in the computational basis.

The parameters of the circuit are optimized using a classical optimizer based on a loss function derived from the measurement outcomes, forming a hybrid quantum-classical learning loop. This structure allows the VQC to act as a quantum analogue of a classical neural network and has been applied to various supervised learning tasks such as binary and multi-class classification Schuld & Killoran (2019); Havlíček et al. (2019). In practice, VQCs are trained on labeled datasets to minimize classification error and output a prediction by measuring specific qubits.

An example of a 2-qubit VQC circuit is shown in Fig. 10, which includes input encoding, parameterized rotations, and entanglement via a CNOT gate.

EXAMPLE OF VQC CIRCUIT FAILS TO SATISFY DIFFERENTIAL PRIVACY

**Example 2 (VQC circuit fails to satisfy differential privacy)** *We investigate whether the 2-qubit Variational Quantum Classifier (VQC) circuit shown in Fig. 10 satisfies the definition of differential privacy for quantum measurements (Definition 4). The circuit consists of input encoding via $R_Y(x_i)$ rotations, trainable gates $R_Z(\theta)$ and $R_Y(\theta)$, and an entangling layer implemented via a CNOT gate.*

*Let $\mathcal{M}_{\text{VQC}}$ denote the computational basis measurement applied to the output of this circuit. We consider two neighboring classical inputs $x = (0,0)$ and $x' = (0,\pi)$, which differ only in the second component. These are encoded into quantum states as $\rho = R_Y(0) \otimes R_Y(0) |00\rangle = |00\rangle$ and $\sigma = R_Y(0) \otimes R_Y(\pi) |00\rangle = |01\rangle$.*

*We apply the following fixed parameters to the trainable layers:*

$$\theta_1 = 0, \quad \theta_2 = 0, \quad \theta_3 = \frac{\pi}{2}, \quad \theta_4 = 0.$$

*Under these parameters, the output probability distributions over the basis states $\{00, 01, 10, 11\}$ are approximately:*

$$\mathcal{M}_{\text{VQC}}(\rho) \approx (0.5, 0.5, 0, 0),$$
$$\mathcal{M}_{\text{VQC}}(\sigma) \approx (0, 0.5, 0.5, 0).$$

*For outcome $00$, $\rho$ yields probability $0.5$ while $\sigma$ yields $0$, leading to an unbounded probability ratio. Thus, $\mathcal{M}_{\text{VQC}}$ does not satisfy $\epsilon$-differential privacy for any finite $\epsilon$. Even under $(\epsilon, \delta)$-differential privacy, it would require $\delta \geq 0.5$ to satisfy the inequality, violating the condition $\delta < 1$.*

*This example demonstrates that quantum measurements in VQC circuits, despite involving randomness, do not inherently satisfy differential privacy and must be carefully modified to ensure privacy guarantees.*

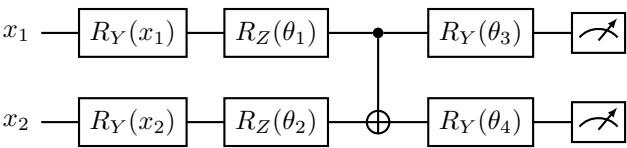

Figure 10: A 2-qubit variational quantum classifier (VQC) circuit. The input $(x_1, x_2)$ is encoded via $R_Y$ rotations, followed by parameterized gates and entangling operations.

EXAMPLE OF MBEM

**Example 3 (Applying MBEM to the VQC measurement)** *We consider the 2-qubit VQC circuit and the quantum measurement $\mathcal{M}_{\mathrm{VQC}}$ as described in Example 2 and Fig. 10. For the input quantum state $\rho = |00\rangle \langle 00|$, assume the original output distribution over $\{00, 01, 10, 11\}$ is approximately $(0.5, 0.5, 0, 0)$.*

*To achieve differential privacy, we apply the Measurement-Based Exponential Mechanism (MBEM) to generate a new output distribution. We define a utility function $u$ such that $u(\rho, 00) = u(\rho, 01) = 0.5$ (the outcomes with non-zero original probabilities), and $u(\rho, i) = 0$ for $i \in \{10, 11\}$. The sensitivity $\Delta u$ is conservatively set to 1.*

*Using Eq. 1, the new output probabilities under MBEM are:*

$$P_{\mathrm{EM}}(00 \mid \rho) = P_{\mathrm{EM}}(01 \mid \rho) = \frac{\exp\left(\frac{\epsilon}{4}\right)}{2\exp\left(\frac{\epsilon}{4}\right) + 2},$$

$$P_{\mathrm{EM}}(10 \mid \rho) = P_{\mathrm{EM}}(11 \mid \rho) = \frac{1}{2\exp\left(\frac{\epsilon}{4}\right) + 2}.$$

*This exponential mechanism redistributes the measurement outcome probabilities in a privacy-preserving way while maintaining high utility on likely outcomes. As a result, even if an adversary has knowledge of the circuit and input encoding, the post-processed measurement outcomes no longer reveal the input state with certainty.*

*This process can be applied to other quantum states similarly. In summary, MBEM enables the measurement $\mathcal{M}_{\mathrm{VQC}}$ to satisfy $\epsilon$-differential privacy, as formally guaranteed by Theorem 4.*

Our MBEM thus provides a practical instance of classical exponential mechanisms adapted to quantum measurement outputs. In hybrid quantum-classical algorithms, it helps ensure both privacy and accuracy by perturbing only the final measurement stage, while preserving the quantum computational process.

EXAMPLE OF DEPOLARIZING NOISE

**Example 4 (Applying depolarizing noise to the VQC measurement)** *We revisit the VQC quantum measurement $\mathcal{M}_{\mathrm{VQC}}$ described in Example 2, using the same local operation-based neighboring relationship and input quantum states $\rho = |00\rangle \langle 00|$ and $\sigma = |01\rangle \langle 01|$ on a 2-qubit system (Dim = 4).*

*To evaluate the privacy of the noisy measurement $\mathcal{M}_{\mathcal{E}_{\mathrm{VQC}}}$ under depolarizing noise with probability $p = 1/3$ and neighboring parameter $\eta = 1$, we apply Theorem 6. The central quantity is $\theta_S$, which depends on the spectral properties of the coarse-grained measurement operator $\Pi_S = \sum_{k \in S} \Pi_{\mathrm{VQC},k}$ for all $S \subseteq \{00, 01, 10, 11\}$.*

*Assume (from simulation or analysis of the VQC structure) that for any single outcome $i \in \{00, 01, 10, 11\}$, the measurement operators satisfy: $\lambda_{\max}(\Pi_{\mathrm{VQC},i}) = \frac{1}{2}$, $\lambda_{\min}(\Pi_{\mathrm{VQC},i}) = 0$, and $\mathrm{tr}(\Pi_{\mathrm{VQC},i}) = 1$. Using Eq. 2, we obtain:*

$$\theta_S = \frac{4(1-p) \cdot \lambda_{\max}(\Pi_S) + p \cdot \mathrm{tr}(\Pi_S)}{p \cdot \mathrm{tr}(\Pi_S)} = \frac{4 \cdot \frac{2}{3} \cdot \frac{1}{2} + \frac{1}{3}}{\frac{1}{3}} = 5.$$

*For larger subsets $|S| \geq 2$, since $0 \leq \lambda_{\min}(\Pi_S) \leq \lambda_{\max}(\Pi_S) \leq 1$ and $\mathrm{tr}(\Pi_S) = |S|$, we derive:*

$$\theta_S \leq \frac{4(1-p) \cdot 1 + p \cdot |S|}{p \cdot |S|} = \frac{\frac{8}{3} + \frac{1}{3}|S|}{\frac{1}{3}|S|} \leq 5.$$

*Therefore, the worst-case value is $\max_{S \subseteq \{00,01,10,11\}} \theta_S = 5$, and the corresponding differential privacy guarantee is $\bar{\epsilon} = \ln 5$.*

*This result demonstrates that although the original measurement $\mathcal{M}_{\mathrm{VQC}}$ does not satisfy differential privacy, the addition of depolarizing noise yields a noisy measurement that satisfies $\ln(5)$-differential privacy for quantum measurements.*

## C.2 Differential privacy for GHZ circuit

**Example 5** *We have selected a 3-qubit quantum circuit known as the GHZ circuit, which is designed to produce a GHZ state. These states find important applications in quantum computing and communication, serving purposes like quantum error correction and secure quantum key distribution Greenberger et al. (1989); Mermin (1990); Caves et al. (2002). The specific configuration of the circuit is*

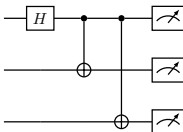

*After the quantum circuit, we employ the quantum measurement $\mathcal{M} = \{\Pi_i\}_{0 \le i \le 7}$ to retrieve the classical information:*

$$
\begin{aligned}
\Pi_0 &= |000\rangle\langle000|, & \Pi_1 &= |001\rangle\langle001|, \\
\Pi_2 &= |010\rangle\langle010|, & \Pi_3 &= |011\rangle\langle011|, \\
\Pi_4 &= |100\rangle\langle100|, & \Pi_5 &= |101\rangle\langle101|, \\
\Pi_6 &= |110\rangle\langle110|, & \Pi_7 &= |111\rangle\langle111|.
\end{aligned}
$$

*As stated earlier in the Heisenberg picture, the GHZ circuit and the quantum measurement can be considered together as a new quantum measurement. We can derive the measurement as $\mathcal{M}_{GHZ} = \{\Pi_{GHZ,i}\}_{0 \le i \le 7}$ with*

$$
\begin{aligned}
\Pi_{GHZ,0} &= \tfrac{1}{2}|000\rangle\langle000| + \tfrac{1}{2}|100\rangle\langle100|, \\
\Pi_{GHZ,1} &= \tfrac{1}{2}|001\rangle\langle001| + \tfrac{1}{2}|101\rangle\langle101|, \\
\Pi_{GHZ,2} &= \tfrac{1}{2}|010\rangle\langle010| + \tfrac{1}{2}|110\rangle\langle110|, \\
\Pi_{GHZ,3} &= \tfrac{1}{2}|011\rangle\langle011| + \tfrac{1}{2}|111\rangle\langle111|, \\
\Pi_{GHZ,4} &= \tfrac{1}{2}|011\rangle\langle011| + \tfrac{1}{2}|111\rangle\langle111|, \\
\Pi_{GHZ,5} &= \tfrac{1}{2}|010\rangle\langle010| + \tfrac{1}{2}|110\rangle\langle110|, \\
\Pi_{GHZ,6} &= \tfrac{1}{2}|001\rangle\langle001| + \tfrac{1}{2}|101\rangle\langle101|, \\
\Pi_{GHZ,7} &= \tfrac{1}{2}|000\rangle\langle000| + \tfrac{1}{2}|100\rangle\langle100|.
\end{aligned}
$$

**Example 6 (Continuing Example 5)** *We investigate the DP budgets offered by the quantum measurement $\mathcal{M}_{GHZ}$ as illustrated in Example 5, employing the local operation-based neighboring relationship for quantum states $\rho \sim \sigma$. Consider two neighboring quantum states $\rho = |000\rangle\langle000|$ and $\sigma = |001\rangle\langle001|$. The probability distributions of measuring the two states by $\mathcal{M}_{GHZ}$ over the outcome set $\{0, 1, \ldots, 7\}$ are $(\tfrac{1}{2}, 0, 0, 0, 0, 0, 0, \tfrac{1}{2})$ and $(0, \tfrac{1}{2}, 0, 0, 0, 0, \tfrac{1}{2}, 0)$.*

*As per the definition of DP in Definition 4, the probabilities for each outcome should be relatively similar. When considering $\epsilon$-DP, it is observed that for the outcome 0, the measurement $\mathcal{M}_{GHZ}$ fails to meet the differential privacy requirement regardless of the chosen magnitude of $\epsilon$. Moreover, in the context of $(\epsilon, \delta)$-DP, it necessitates $\delta$ to be 1 violating the constraint $\delta < 1$. Irrespective of the type of DP considered, it is evident that this quantum measurement lacks differential privacy protection attributes.*

**Example 7 (Continuing Example 5)** *We examine quantum measurement $\mathcal{M}_{GHZ}$ as detailed in Example 5. For quantum state $\rho_{\vec{v}} = |000\rangle\langle000|$, $\mathcal{M}_{GHZ}$ results in outcomes $0, 1, \ldots, 7$ with corresponding probabilities of $(\tfrac{1}{2}, 0, 0, 0, 0, 0, 0, \tfrac{1}{2})$. In the subsequent steps, our objective is to employ the MBEM to compute a new outcome distribution and randomly choose the measurement result to ensure privacy preservation.*

*The utility function $u$ is defined as $u(\rho_{\vec{\nu}}, 0) = u(\rho_{\vec{\nu}}, 7) = \frac{1}{2}$ and $u(\rho_{\vec{\nu}}, i) = 0$ for $1 \leq i \leq 6$. Sensitivity $\Delta u$ is set to 1. Then, using Eq. 1, the distribution of selecting outcomes is*

$$P_{EM}(0|\rho_{\vec{\nu}}) = P_{EM}(7|\rho_{\vec{\nu}}) = \frac{\exp(\frac{\epsilon}{4})}{2\exp(\frac{\epsilon}{4}) + 6},$$

$$P_{EM}(1|\rho_{\vec{\nu}}) = \cdots = P_{EM}(6|\rho_{\vec{\nu}}) = \frac{1}{2\exp(\frac{\epsilon}{4}) + 6}.$$

*Here, we have illustrated how to add noise to quantum state $\rho_{\vec{\nu}}$. This same process can be used for other input states as well. In conclusion, the quantum measurement $\mathcal{M}_{GHZ}$ using our MBEM adheres to $\epsilon$-DP as proven in Theorem 4.*

**Example 8 (Continuing Example 5)** *We employ the quantum measurement $\mathcal{M}_{GHZ}$ as outlined in Example 5, maintaining the same local operation-based neighboring relationship and input states $\rho = |000\rangle\langle 000|$ and $\sigma = |001\rangle\langle 001|$ from Example 6 on a 3-qubit system (Dim = 8).*

*To find the optimal $\bar{\epsilon}$ given by noisy measurement $\mathcal{M}_{\mathcal{E}_{GHZ}}$ with a noisy probability $p = 1/3$ and $\eta = 1$, according to Theorem 6, we first determine the maximum and minimum eigenvalues of $\Pi_S = \sum_{k \in S} \Pi_{GHZ,k}$, for all $S \subseteq \{0, ..., 7\}$. Let $|S|$ indicate the number of elements in the set.*

*For $|S| = 1$ and $i \in \{0, \ldots, 7\}$, we have $\lambda_{\max}(\Pi_{GHZ,i}) = \frac{1}{2}$, $\lambda_{\min}(\Pi_{GHZ,i}) = 0$, $\mathrm{tr}(\Pi_{GHZ,i}) = 1$, and $\theta_S = 9$ by Eq. 2.*

*For $|S| \geq 2$ and any $S \subseteq \{0, ..., 7\}$, we have $\mathrm{tr}(\Pi_S) = |S|$. Combining the fact $0 \leq \lambda_{\min}(\Pi_S) \leq \lambda_{\max}(\Pi_S) \leq 1$, we obtain*

$$\theta_S \leq \frac{8 \cdot (1 - \frac{1}{3}) \cdot 1 + \frac{1}{3} \cdot |S|}{\frac{1}{3} \cdot |S|} \leq 9.$$

*Therefore, $\max_{S \subseteq \{0, ..., 7\}} \theta_S = 9$, and $\bar{\epsilon} = \ln 9$.*

## D POST-PROCESSING AND COMPOSITION THEOREM

As we mentioned in the introduction, with the rapid development of quantum computing and the superior computational capabilities of quantum computers, an increasing number of classical problems are now being addressed using hybrid quantum-classical algorithms denoted as $\mathcal{A} = (\mathcal{E}, \mathcal{M})$. The process involves converting a classical dataset $\vec{\nu}$ into a quantum input state $\rho_{\vec{\nu}}$ through quantum encoding, as depicted in Fig. 3 of Section 2. Subsequently, the quantum circuit $\mathcal{E}$ and quantum measurement $\mathcal{M}$ are applied, leading to the retrieval of classical data (measurement outcome set $\mathcal{O}$) based on a probability distribution determined by the measurement $\mathcal{M}$. This approach serves as a model for leveraging hybrid quantum-classical algorithms to solve classical problems. In the context of privacy considerations, a pertinent question arises: Does the post-process theorem still apply when employing quantum computing for classical problem-solving? Encouragingly, favorable outcomes can be observed with the use of neighboring-preserving quantum encoding techniques and noting that we can use measurement $\mathcal{M}_{\mathcal{E}}$ to describe the above evolution of hybrid quantum-classical algorithm $\mathcal{A} = (\mathcal{E}, \mathcal{M})$ in the Heisenberg picture in Section 2.

**Corollary 2** *Suppose $\Lambda$ represents a neighboring-preserving quantum encoding. If $\mathcal{M} = \{\Pi_i\}_{i \in \mathcal{O}}$ is a quantum measurement that is $(\epsilon, \delta)$-DP, then $\mathcal{K} \circ \mathcal{M} \circ \Lambda$ is $(\epsilon, \delta)$-CDP for any randomized function $\mathcal{K} : \mathcal{O} \to \mathcal{O}'$.*

In the result above, $\mathcal{M} \circ \Lambda$ represents a classical randomized function with classical input, classical output, and quantum processing. This indicates that integrating a differentially private quantum measurement into a classical problem can also provide privacy protection against any subsequent post-processing analysis. Moreover, when combined with the second assertion in Theorem 1, our DP can deliver efficient privacy safeguarding in CDP that can withstand post-process attacks. This improves the privacy protection when utilizing quantum computers for solving classical problems.

## D.1 COMPOSITION THEOREM

In classical computing, it is crucial to explore how two algorithms can be combined, as this approach naturally leads to the creation of more advanced algorithms. Researchers have introduced composition theorems for classical and quantum differential privacy to safeguard the privacy of such combined algorithms Dwork et al. (2014); Guan et al. (2023). Similarly, in our DP framework of quantum measurement, we aim to investigate whether the fusion of two differentially private quantum measurements maintains its differentially private nature. To accomplish this goal, we must initially introduce how two quantum measurements can be combined and subsequently formulate the relevant composition theorem by extending the relationships between neighboring states from a single system to those within composed systems.

Given two quantum measurements, $\mathcal{M}_1 = \{\Pi_i\}_{i \in \mathcal{O}_1}$ and $\mathcal{M}_2 = \{\Pi_j\}_{j \in \mathcal{O}_2}$, their joint measurement can be expressed as

$$\mathcal{M}_{1,2} = \{\Pi_{i,j} = \Pi_i \otimes \Pi_j, i \in \mathcal{O}_1, j \in \mathcal{O}_2\}.$$

The combination $\mathcal{M}_{1,2}$ is viewed as a randomized function from the tensor product of the state space $\mathcal{D}(\mathcal{H}_1) \otimes \mathcal{D}(\mathcal{H}_2)$ to a probability distribution over the set of joint measurement outcomes $\mathcal{O}_1 \times \mathcal{O}_2$.

In the composed state space $\mathcal{D}(\mathcal{H}_1) \otimes \mathcal{D}(\mathcal{H}_2)$, quantum states are in the form of tensor products, such as $\rho_{1,2} = \rho_1 \otimes \rho_2$, where $\rho_i \in \mathcal{D}(\mathcal{H}_i)$ for $i \in \{0, 1\}$. This type of state $\rho_{1,2}$ is referred to as a *product state*. Two product states, $\rho_{1,2} = \rho_1 \otimes \rho_2$ and $\sigma_{1,2} = \sigma_1 \otimes \sigma_2$, are considered neighboring, denoted as $\rho_{1,2} \sim \sigma_{1,2}$, when the respective subsystem states are also neighboring, meaning $\rho_i \sim \sigma_i$ for $i \in \{1, 2\}$ Zhou & Ying (2017). This notion of neighboring relationships can be straightforwardly generalized to any finite composite state space $\mathcal{D}(\mathcal{H}_1) \otimes \mathcal{D}(\mathcal{H}_2) \otimes \cdots \otimes \mathcal{D}(\mathcal{H}_n)$.

Now, we present a composition theorem for differentially private quantum measurements within our framework.

**Theorem 7** *If quantum measurements $\mathcal{M}_1 = \{\Pi_i\}_{i \in \mathcal{O}_1}$ and $\mathcal{M}_2 = \{\Pi_j\}_{j \in \mathcal{O}_2}$ are $(\epsilon_1, \delta_1)$-DP and $(\epsilon_2, \delta_2)$-DP on Hilbert spaces $\mathcal{H}_1$ and $\mathcal{H}_2$ respectively, then the combined measurement $\mathcal{M}_{1,2}$ is $(\epsilon_1 + \epsilon_2, \delta_1 + \delta_2)$-DP.*

Compared to the composition theorem in QDP Hirche et al. (2023), which states that the composition satisfies $(\epsilon_1 + \epsilon_2, \bar{\delta})$-differential privacy with $\bar{\delta} = \min\{\delta_1 + e^{\epsilon_1}\delta_2, e^{\epsilon_2}\delta_1 + \delta_2\}$, our composition theorem achieves a tighter bound. Specifically, we show that the composition satisfies $(\epsilon_1 + \epsilon_2, \delta_1 + \delta_2)$-DP. This result aligns the composition bound in our framework with the well-established CDP composition bound, significantly simplifying the analysis and improving the privacy guarantee compared to QDP. Thus, our composition theorem highlights the advantage of differential privacy framwork of quantum measurement in bridging classical and quantum differential privacy while offering a more efficient composition bound.

This result can be extended to the finite composition case.

**Corollary 3** *For each $k \in [n] = \{1, 2, \ldots, n\}$, suppose $\mathcal{M}_k = \{\Pi_{i_k}\}_{i_k \in \mathcal{O}_k}$ is an $(\epsilon_k, \delta_k)$-DP quantum measurement on the Hilbert space $\mathcal{H}_k$. Then their joint measurement $\mathcal{M}_{[n]}$ is $(\sum_{i=1}^{n} \epsilon_i, \sum_{i=1}^{n} \delta_i)$-DP, where*

$$\mathcal{M}_{[n]} = \left\{ \Pi_{i_1,\ldots,i_n} = \bigotimes_{k=1}^{n} \Pi_{i_k} : (i_1, \ldots, i_n) \in \mathcal{O}_1 \times \cdots \times \mathcal{O}_n \right\}.$$

Given the above result along with the post-processing theorem, our method for differentially private quantum measurements proves to be successful. As a result, we can now focus on developing such measurements to protect the privacy of hybrid quantum-classical algorithms.

## E PROOFS OF THEORETICAL RESULTS

### E.1 PROOF OF THEOREM 1

**Proof 1** *The first claim of the theorem can be deduced from the definitions of differential privacy of quantum measurement and quantum differential privacy.*

*Proving the second assertion of the theorem: It is easy to find that $Range(\mathcal{M} \circ \Lambda) = \mathcal{O}$. For any two neighboring classical input states $\vec{\nu} \sim \vec{\omega}$, since $\Lambda$ is neighboring-preserving, $\Lambda(\vec{\nu}) \sim \Lambda(\vec{\omega})$. As $\mathcal{M}$ is $(\epsilon, \delta)$-DP, we have*

$$\text{tr}(\Pi_S(\Lambda(\vec{\nu}))) \le e^\epsilon \text{tr}(\Pi_S(\Lambda(\vec{\omega}))) + \delta,$$

*for all $S \subseteq \mathcal{O}$. Then we have*

$$
\begin{aligned}
\Pr[\mathcal{M} \circ \Lambda(\vec{\nu}) \in S] &= \text{tr}(\Pi_S(\Lambda(\vec{\nu}))) \\
&\le e^\epsilon \text{tr}(\Pi_S(\Lambda(\vec{\omega}))) + \delta \\
&= e^\epsilon \Pr[\mathcal{M} \circ \Lambda(\vec{\omega}) \in S] + \delta,
\end{aligned}
$$

*for all $S \subseteq \mathcal{O}$. That means $\mathcal{M} \circ \Lambda$ is $(\epsilon, \delta)$-CDP.*

### E.2 PROOF OF THEOREM 2

This follows from the post-process theorem in classical differential privacy (Dwork et al., 2014, Proposition 2.1, Remark 3.1) by noting that $\mathcal{M}$ is a randomized function over a classical set. Specifically, consider a randomized function $\mathcal{K} : \mathcal{O} \to \mathcal{O}'$. Observe that

$$D_\infty^\delta(\mathcal{K}(P)\|\mathcal{K}(Q)) \le D_\infty^\delta(P\|Q).$$

It means that if $\mathcal{M}$ is $(\epsilon, \delta)$-differentially privacy, so is $\mathcal{K} \circ \mathcal{M}$. Here $D_\infty^\delta(P\|Q)$ stands for the $\delta$-max divergence of random variables $P$ and $Q$, the specific definition is as follows:

$$D_\infty^\delta(P\|Q) = \max_{S \subseteq \mathcal{O}; \Pr[P \in S] \ge \delta} \ln \frac{\Pr[P \in S] - \delta}{\Pr[Q \in S]}.$$

### E.3 PROOF OF THEOREM 4

**Proof 2** *For all $\rho \sim \sigma$ and $i \in \mathcal{O}$, we have*

$$
\begin{aligned}
\frac{P_{EM}(i|\rho)}{P_{EM}(i|\sigma)} &= \frac{\left( \frac{\exp\left(\frac{\epsilon u(\rho,i)}{2\Delta u}\right)}{\sum_{j \in \mathcal{O}} \exp\left(\frac{\epsilon u(\rho,j)}{2\Delta u}\right)} \right)}{\left( \frac{\exp\left(\frac{\epsilon u(\sigma,i)}{2\Delta u}\right)}{\sum_{j \in \mathcal{O}} \exp\left(\frac{\epsilon u(\sigma,j)}{2\Delta u}\right)} \right)} \\
&= \left( \frac{\exp\left(\frac{\epsilon u(\rho,i)}{2\Delta u}\right)}{\exp\left(\frac{\epsilon u(\sigma,i)}{2\Delta u}\right)} \right) \cdot \left( \frac{\sum_{j \in \mathcal{O}} \exp\left(\frac{\epsilon u(\sigma,j)}{2\Delta u}\right)}{\sum_{j \in \mathcal{O}} \exp\left(\frac{\epsilon u(\rho,j)}{2\Delta u}\right)} \right) \\
&= \exp\left( \frac{\epsilon(u(\rho,i) - u(\sigma,i))}{2\Delta u} \right) \\
&\quad \cdot \left( \frac{\sum_{j \in \mathcal{O}} \exp\left(\frac{\epsilon u(\sigma,j)}{2\Delta u}\right)}{\sum_{j \in \mathcal{O}} \exp\left(\frac{\epsilon u(\rho,j)}{2\Delta u}\right)} \right) \\
&\le \exp(\frac{\epsilon}{2}) \exp(\frac{\epsilon}{2}) \cdot \left( \frac{\sum_{j \in \mathcal{O}} \exp\left(\frac{\epsilon u(\rho,j)}{2\Delta u}\right)}{\sum_{j \in \mathcal{O}} \exp\left(\frac{\epsilon u(\rho,j)}{2\Delta u}\right)} \right) \\
&= \exp(\epsilon).
\end{aligned}
$$

*Similarly, by the symmetry, we obtain that*

$$\frac{P_{EM}(i|\rho)}{P_{EM}(i|\sigma)} \ge \exp(-\epsilon).$$

### E.4 PROOF OF THEOREM 5

**Proof 3** *According to Theorem 3.5 in Guan et al. (2023), by choosing $\mathcal{U}$ as $I$, we can arrive at this conclusion.*

## E.5 PROOF OF THEOREM 6

**Proof 4** *According to the definition of the depolarizing noise, we have that, for any matrix A,*

$$\mathcal{E}^{\dagger}_{Dep}(A) = (1-p)A + \text{tr}(A)\frac{p}{Dim}I.$$

*So we can get that*

$$
\begin{aligned}
\sum_{k \in S} \mathcal{E}^{\dagger}_{Dep}(\Pi_k) &= \mathcal{E}^{\dagger}_{Dep}(\sum_{k \in S} \Pi_k) = \mathcal{E}^{\dagger}_{Dep}(\Pi_S) \\
&= (1-p)\Pi_S + \text{tr}(\Pi_S)\frac{p}{Dim}I.
\end{aligned}
$$

*Susequently, we have*

$$\lambda_{\max}(\mathcal{E}^{\dagger}_{Dep}(\Pi_S)) = (1-p)\lambda_{\max}(\Pi_S) + \text{tr}(\Pi_S)\tfrac{p}{Dim},$$
$$\lambda_{\min}(\mathcal{E}^{\dagger}_{Dep}(\Pi_S)) = (1-p)\lambda_{\min}(\Pi_S) + \text{tr}(\Pi_S)\tfrac{p}{Dim}.$$

*Then by using Proposition 1, we can get the conclusion.*

## E.6 PROOF OF COROLLARY 1

**Proof 5** *For any quantum measurement $\mathcal{M} = \{\Pi_i\}_{i \in \mathcal{O}}$ and for any $S \subseteq \mathcal{O}$, we have $0 \leq \lambda_{\min}(\Pi_s) \leq \lambda_{\max}(\Pi_S) \leq \text{tr}(\Pi_S)$. Then according to Theorem 6*

$$
\begin{aligned}
\theta_S &= \frac{Dim \cdot (1-p)\lambda_{\max}(\Pi_S) + \text{tr}(\Pi_S)p}{Dim \cdot (1-p)\lambda_{\min}(\Pi_S) + \text{tr}(\Pi_S)p} \\
&\leq \frac{Dim \cdot (1-p)\text{tr}(\Pi_S) + \text{tr}(\Pi_S)p}{\text{tr}(\Pi_S)p} \\
&= \frac{Dim \cdot (1-p) + p}{p}.
\end{aligned}
$$

*So $\bar{\theta} = \max_{S \subseteq \mathcal{O}} \theta_S \leq \frac{Dim \cdot (1-p)+p}{p}$ and*

$$
\begin{aligned}
\bar{\epsilon} &\leq \ln[(\frac{Dim \cdot (1-p) + p}{p} - 1)\eta + 1] \\
&= \ln\left(\frac{Dim \cdot (1-p)}{p}\eta + 1\right).
\end{aligned}
$$

*Which means $\mathcal{M}_{\mathcal{E}_{Dep}}$ is $\epsilon$-DP, where*

$$\epsilon = \ln\left(\frac{Dim \cdot (1-p)}{p}\eta + 1\right).$$

## E.7 PROOF OF THEOREM 7

To prove the composition theorem, we need the following lemma.

**Lemma 1** *Let $\mu_1, \nu_1$ be distributions over a finite set $K_1$, and let $\mu_2, \nu_2$ be distributions over a finite set $K_2$, such that:*
*For any $S_1 \subseteq K_1$,*

$$\mu_1(S_1) \leq e^{\epsilon_1} \cdot \nu_1(S_1) + \delta_1.$$

*For any $S_2 \subseteq K_2$,*

$$\mu_2(S_2) \leq e^{\epsilon_2} \cdot \nu_2(S_2) + \delta_2.$$

*Here $\mu_i(S_i) = \sum_{s \in S_i} \mu_i(s)$ and $\nu_i(S_i) = \sum_{s \in S_i} \nu_i(s)$, for $i \in \{1, 2\}$.*

*Define $\mu_{1,2}$ and $\nu_{1,2}$ be the distributions over set $K_1 \times K_2$, when $s = (k_1, k_2) \in K_1 \times K_2$,*

$$\mu_{1,2}(s) \triangleq \mu_1(k_1) \cdot \mu_2(k_2), \ \nu_{1,2}(s) \triangleq \nu_1(k_1) \cdot \nu_2(k_2).$$

*Then, for any $S \subseteq K_1 \times K_2$,*

$$\mu_{1,2}(S) \leq e^{\epsilon_1 + \epsilon_2} \nu_{1,2}(S) + \delta_1 + \delta_2.$$

*Here $\mu_{1,2}(S) = \sum_{s \in S} \mu_{1,2}(s)$ and $\nu_{1,2}(S) = \sum_{s \in S} \nu_{1,2}(s)$.*

**Proof 6** *First, we separate $K_1$ into two parts $A_1, A_2$, satisfying $A_1 \cup A_2 = K_1, A_1 \cap A_2 = \emptyset$, for any $a \in A_1$, $\mu_1(a) > e^{\epsilon_1} \nu_1(a)$, and for any $a \in A_2$, $\mu_1(a) \leq e^{\epsilon_1} \nu_1(a)$. We separate $K_2$ into two parts like this: $B_1 \cup B_2 = K_2, B_1 \cap B_2 = \emptyset$, for any $b \in B_1$, $\mu_2(b) > e^{\epsilon_2} \nu_2(b)$, and for any $b \in B_2$, $\mu_2(b) \leq e^{\epsilon_2} \nu_2(b)$. Then we separate $S$ into three parts $C_1 = S \cap (A_1 \times K_2), C_2 = (S - C_1) \cap (K_1 \times B_1), C_3 = S - C_1 - C_2$.*

*Let $C_a = (a \times K_2) \cap S, a \in K_1, C_b = (K_1 \times b) \cap (S - C_1), b \in K_2$. For all $a \in A_1$, it is easy to prove that $\mu_{1,2}(C_a) - e^{\epsilon_1 + \epsilon_2} \nu_{1,2}(C_a) \leq \mu_1(a)\delta_2$. Furthermore, for all $b \in B_1$, it is easy to prove that $\mu_{1,2}(C_b) - e^{\epsilon_1 + \epsilon_2} \nu_{1,2}(C_b) \leq \mu_2(b)\delta_1$. For any $c \in C_3$, it is easy to prove that $\mu_{1,2}(c) - e^{\epsilon_1 + \epsilon_2} \nu_{1,2}(c) \leq 0$. Then we have*

$$
\begin{aligned}
& \mu_{1,2}(S) - e^{\epsilon_1 + \epsilon_2} \nu_{1,2}(S) \\
= & \bigcup_{a \in A_1} \mu_{1,2}(C_a) + \bigcup_{b \in B_1} \mu_{1,2}(C_b) + \bigcup_{c \in C_3} \mu_{1,2}(c) \\
& - e^{\epsilon_1 + \epsilon_2} \left( \bigcup_{a \in A_1} \nu_{1,2}(C_a) + \bigcup_{b \in B_1} \nu_{1,2}(C_b) + \bigcup_{c \in C_3} \nu_{1,2}(c) \right) \\
\leq & \bigcup_{a \in A_1} \mu_1(a)\delta_2 + \bigcup_{b \in B_2} \mu_2(b)\delta_1 \\
\leq & \delta_1 + \delta_2.
\end{aligned}
$$

Now we can prove Theorem 7.

**Proof 7** *When $i \in \mathcal{O}_1, j \in \mathcal{O}_2$,*

$$
\begin{aligned}
\mathrm{tr}(\Pi_{i,j} \cdot (\rho \otimes \sigma)) &= \mathrm{tr}((\Pi_i \otimes \Pi_j)(\rho \otimes \sigma)) \\
&= \mathrm{tr}(\Pi_i \rho) \times \mathrm{tr}(\Pi_j \sigma).
\end{aligned}
$$

*According to lemma 1, for any $S \subseteq \mathcal{O}_1 \times \mathcal{O}_2$, we have*

$$
\begin{aligned}
& \sum_{(i,j) \in S} \mathrm{tr}(\Pi_{i,j} \cdot (\rho \otimes \sigma)) \\
\leq & \; e^{(\epsilon_1 + \epsilon_2)} \cdot \sum_{(i,j) \in S} \mathrm{tr}(\Pi_{i,j} \cdot (\rho' \otimes \sigma')) + \delta_1 + \delta_2.
\end{aligned}
$$

*So the combination $\mathcal{M}_{1,2}$ is $(\epsilon_1 + \epsilon_2, \delta_1 + \delta_2)$-differentially private.*

## F    EXPONENTIAL MECHANISM

The exponential mechanism is a method used to select the "best" output while ensuring differential privacy, particularly when directly adding noise to the computed result would destroy its value. The exponential mechanism uses a utility function to define the probability distribution of the outputs, thus choosing outputs with higher utility scores while ensuring privacy.

### F.1    STEPS OF THE EXPONENTIAL MECHANISM

1. **Define Utility Function**: First, define a utility function $u : \mathbb{N}^{|X|} \times R \to \mathbb{R}$, which maps a dataset $x$ and output $r$ to a utility score. The utility function represents the quality or benefit of each possible output.

2. **Compute Utility Scores**: For a given dataset $x$ and all possible outputs $r \in R$, compute the utility score $u(x, r)$ for each $r$.

3. **Compute Probability Distribution**: Assign a probability to each possible output based on its utility score. Specifically, the probability of output $r$ being chosen is proportional to $\exp\left(\frac{\epsilon u(x,r)}{2\Delta u}\right)$, where $\Delta u$ is the sensitivity of the utility function, defined as:

$$\Delta u = \max_{r \in R, x, x'} |u(x,r) - u(x',r)|$$

Here, $x$ and $x'$ are any two neighboring datasets.

4. **Normalization**: Normalize these probabilities so that they sum to 1. Specifically, the probability of choosing output $r$ is:

$$P(r|x) = \frac{\exp\left(\frac{\epsilon u(x,r)}{2\Delta u}\right)}{\sum_{r' \in R} \exp\left(\frac{\epsilon u(x,r')}{2\Delta u}\right)}$$

5. **Sampling**: Sample an output from the set of possible outputs $R$ according to the computed probability distribution.

### F.2 INTUITIVE EXPLANATION

The core idea of the exponential mechanism is to choose outputs with higher utility scores more frequently by assigning them higher probabilities. This ensures that, even with the addition of noise to protect privacy, an output close to the optimal can still be selected. As the utility score decreases, the probability of selection decreases exponentially, ensuring that high-utility outputs have a greater chance of being chosen than low-utility ones.

- There are five bidders, each bidding a different amount for an item: $1.00, $2.00, $3.00, $4.00, and $5.00.

- The goal is to choose a price that maximizes the total revenue.

- Define the utility function $u(x,p)$ as the total revenue at price $p$, where $x$ represents all bids.

- Use the exponential mechanism to select a price based on the utility function.

Suppose we have a dataset of bids: $x = [1, 2, 3, 4, 5]$. The utility function $u(x,p)$ computes the total revenue for a given price $p$. For example:

If the price is $1, all five bidders will buy the item, resulting in a total revenue of $5. For higher prices: at $2, the revenue is $8; at $3, the revenue is $9; at $4, the revenue is $8; and at $5, the revenue is $5.

For a given $\epsilon = 1.0$ and a sensitivity $\Delta u = 1$, we compute the probability of selecting each price using the exponential mechanism. The probabilities are proportional to $\exp(\epsilon u(x,p)/2\Delta u)$, let $Z = \exp(2.5) + \exp(4) + \exp(4.5) + \exp(4) + \exp(2.5)$, after normalize these probabilities so that they sum to 1, we get:

$$P(1) = \frac{\exp(2.5)}{Z}, P(2) = \frac{\exp(4)}{Z}, P(3) = \frac{\exp(4.5)}{Z},$$
$$P(4) = \frac{\exp(4)}{Z}, P(5) = \frac{\exp(2.5)}{Z}.$$

Finally, we sample a price from these normalized probabilities. This ensures that prices with higher utility scores (total revenue) have a higher chance of being selected, while maintaining differential privacy.

## G CHARACTERIZING HYBRID DIFFERENTIAL PRIVACY

**Proposition 1** *(Guan et al., 2023, Theorem 4.1) Let $\mathcal{M} = \{\Pi_i\}_{i \in \mathcal{O}}$ represent a quantum measurement. The following conditions hold for $\mathcal{M}$:*

- $\mathcal{M}$ is $(\epsilon, \delta)$-*DP if and only if* $\delta \geq \max_{S \subseteq \mathcal{O}} \delta_S$, *where*

$$\delta_S = \eta \lambda_{\max}(\Pi_S) - (e^\epsilon + \eta - 1)\lambda_{\min}(\Pi_S)$$

  *and the matrix* $\Pi_S = \sum_{k \in S} \Pi_k$. *The terms* $\lambda_{\max}(\Pi_S)$ *and* $\lambda_{\min}(\Pi_S)$ *refer to the maximum and minimum eigenvalues of the positive matrix* $\Pi_S$, *respectively.*

- $\mathcal{M}$ *is said to be* $\epsilon$-*DP if and only if* $\epsilon \geq \epsilon^*$, *where* $\epsilon^*$ *is the optimal bound of* $\epsilon$ *given by*

$$\epsilon^* = \ln[(\kappa^* - 1)\eta + 1] \quad and \quad \kappa^* = \max_{S \subseteq \mathcal{O}} \kappa(\Pi_S),$$

  *Here,* $\kappa(\Pi_S) = \frac{\lambda_{\max}(\Pi_S)}{\lambda_{\min}(\Pi_S)}$ *represents the condition number of matrix* $\Pi_S$, *and if* $\lambda_{\min}(\Pi_S) = 0$, *then* $\kappa(\Pi_S)$ *is considered as* $+\infty$.

## H  UTILITY LOSS EVALUATION

To analyze the trade-off between utility and privacy, we employ the Kullback-Leibler (KL) divergence to evaluate the effectiveness of differentially private mechanisms. KL divergence measures the discrepancy between a model's probability distribution and the actual distribution, and is widely used in classical differential privacy to assess how distinguishable the private output is from the original one Duchi et al. (2013); Ghazi & Issa (2024); Kairouz et al. (2014); Asoodeh & Zhang (2024).

In our context, we compute the maximum KL divergence between the measurement outcome distributions before and after applying a noise mechanism, across all quantum states. Let $\mathcal{M}$ denote the original quantum measurement, and $\mathcal{M}_N$ the measurement after applying a noise mechanism $N$ (e.g., MBEM or depolarizing noise). The utility loss is defined as:

$$\mathrm{UL}(N) = \max_{\rho \in \mathcal{D}(\mathcal{H})} D_{\mathrm{KL}}(\mathcal{M}(\rho) \| \mathcal{M}_N(\rho)).$$

A smaller $\mathrm{UL}(N)$ indicates lower distortion from the noise and thus better utility preservation, whereas a larger value suggests greater utility degradation. This quantitative metric allows us to systematically compare the impact of different noise mechanisms within our framework.

## I  CLASSICAL NOISE-BASED QUANTUM DIFFERENTIAL PRIVACY METHOD

Recent studies have proposed achieving quantum differential privacy by leveraging the addition of classical noise to the outputs of quantum measurements. This method provides privacy guarantees by randomizing the results of measurements on quantum states, thereby making the output distributions for neighboring quantum states statistically indistinguishable. Specifically, the method employs two widely-used classical noise mechanisms:

- **Laplace Mechanism:** By adding noise drawn from a Laplace distribution with a scale parameter $b$, where $b \geq \Delta f / \epsilon$ (with $\Delta f$ representing the sensitivity of the measurement and $\epsilon$ the privacy budget), this mechanism ensures that the measurement output remains $\epsilon$-differentially private.
- **Gaussian Mechanism:** Gaussian noise with variance $\sigma^2 \geq 2\ln(1.25/\delta)\Delta^2/\epsilon^2$ is added to the measurement results, achieving $(\epsilon, \delta)$-differential privacy. Here, $\Delta$ denotes the sensitivity, $\epsilon$ is the privacy budget, and $\delta$ controls the probability of privacy failure.

This hybrid approach integrates classical differential privacy techniques into quantum settings. The central idea is that classical noise mechanisms exploit the statistical properties of quantum measurement outputs while accounting for the bounded trace distance of neighboring quantum states. This combination reduces the distinguishability of outputs derived from neighboring quantum states and provides formal privacy guarantees.

The method is particularly advantageous in noisy intermediate-scale quantum (NISQ) devices, where classical noise mechanisms complement inherent quantum noise. The authors further demonstrate that this technique aligns with a unified framework for quantum and classical differential privacy, ensuring robust privacy protection without significantly compromising utility.

## J  ADDITIONAL EXPERIMENTS OF DIFFERENTIAL PRIVACY MECHANISMS

This section provides supplementary experiments to further evaluate the effectiveness and utility of differential privacy mechanisms of quantum measurement. Specifically, we present detailed analyses and experimental results for two key mechanisms: the Measurement-Based Exponential Mechanism (MBEM) and the Depolarizing Noise mechanism. The goal is to explore how these mechanisms perform under different configurations, with a focus on balancing privacy and utility.

### PLATFORM AND COMPUTE COST

All experiments were conducted on a MacBook Pro equipped with an Apple M1 Pro chip (10-core CPU, 16 GB unified memory, integrated GPU), running macOS Monterey 12.0.1.

To evaluate the computational cost of our methods, we measured the execution time for privacy-utility trade-off experiments using both MBEM and depolarizing noise mechanisms on two types of circuits—random circuits (RC) and variational quantum circuits (VQC)—with 10 and 12 qubits. The results are shown in Table 4.

We observe that MBEM is extremely lightweight, requiring less than 3 seconds even on 12-qubit circuits. In contrast, the depolarizing noise mechanism is more computationally intensive due to spectral analysis over high-dimensional measurement operators. For instance, on 12-qubit VQCs, the depolarizing method takes approximately 10 minutes.

These experiments demonstrate that the computational cost is modest for MBEM and moderate but still manageable for depolarizing noise. All experiments were run on a single consumer-grade laptop without GPU acceleration.

### J.1  ADDITIONAL ANALYSIS OF MBEM FOR HYBRID DIFFERENTIAL PRIVACY

In this subsection, we extend the analysis of the MBEM introduced in Example 7. We explore the application of the MBEM to safeguard the $\epsilon$-DP of a quantum state $\rho_{\vec{\nu}} = |000\rangle \langle 000|$ against any other state when measured by the quantum measurement $\mathcal{M}_{\text{GHZ}}$. Table 2 illustrates the resulting probability distributions with varying levels of noise error introduced by the MBEM to achieve different levels of privacy as defined by $\epsilon$ in our framework. The second row in the table represents the distribution of measurement outcomes without the mechanism, which does not ensure privacy protection ($\epsilon = \infty$). The first column in the table denotes the expected level of differential privacy provided by the MBEM in Theorem 4, while columns two through nine display the adjusted measurement outcome distributions due to the mechanism. The final column indicates the more actual level, denoted by $\epsilon_D$, of differential privacy achieved by the mechanism as a baseline.

The table illustrates that as the value of $\epsilon$ decreases, the MBEM introduces more noise, thereby enhancing privacy protection. This highlights the mechanism's effectiveness in ensuring privacy. However, the level of privacy guaranteed by $\epsilon$ deviates significantly from the true privacy level $\epsilon_D$. *To address this discrepancy and obtain a more accurate estimation of the privacy protection offered by the MBEM, selecting a more suitable value for $\Delta u$, which represents the sensitivity of the mechanism, is crucial.* In Example 3 and Table 2, we assume $\Delta u = 1$, but a more precise value can be calculated using the formula in Eq. ??, resulting in $\Delta u = 1/2$. By substituting this refined value into the example, we derive the outcomes presented in Table 3.

A comparison between Tables 2 and 3 reveals that to achieve the same expected differential privacy effect $\epsilon$, setting $\Delta u$ to 1/2 reduces the added noise, as evidenced by the probability distribution post-noise addition closely resembling the original distribution. Furthermore, a more accurate $\Delta u$ leads to $\epsilon$ approaching $\epsilon_D$, indicating that a precise $\Delta u$ can deliver the desired privacy protection with minimal interference from noise, thus balancing privacy and utility. This trade-off is demonstrated in Fig. 5.

### J.2  ADDITIONAL ANALYSIS OF DEPOLARIZING NOISE MECHANISM FOR HYBRID DIFFERENTIAL PRIVACY

In addition to MBEM, we also examine the performance of the depolarizing noise mechanism in achieving hybrid differential privacy. The depolarizing noise mechanism introduces a noisy probabil-

| $\epsilon$ | $P_{EM}(0\|\rho_{\vec{v}})$ | $P_{EM}(1\|\rho_{\vec{v}})$ | $P_{EM}(2\|\rho_{\vec{v}})$ | $P_{EM}(3\|\rho_{\vec{v}})$ | $P_{EM}(4\|\rho_{\vec{v}})$ | $P_{EM}(5\|\rho_{\vec{v}})$ | $P_{EM}(6\|\rho_{\vec{v}})$ | $P_{EM}(7\|\rho_{\vec{v}})$ | $\epsilon_D$ |
|---|---|---|---|---|---|---|---|---|---|
| $\infty$ | 0.5 | 0 | 0 | 0 | 0 | 0 | 0 | 0.5 | $\infty$ |
| 1 | $\approx 0.1499$ | $\approx 0.1167$ | $\approx 0.1167$ | $\approx 0.1167$ | $\approx 0.1167$ | $\approx 0.1167$ | $\approx 0.1167$ | $\approx 0.1499$ | 0.25 |
| 3 | $\approx 0.2069$ | $\approx 0.0977$ | $\approx 0.0977$ | $\approx 0.0977$ | $\approx 0.0977$ | $\approx 0.0977$ | $\approx 0.0977$ | $\approx 0.2069$ | 0.75 |
| 5 | $\approx 0.2680$ | $\approx 0.0770$ | $\approx 0.0770$ | $\approx 0.0770$ | $\approx 0.0770$ | $\approx 0.0770$ | $\approx 0.0770$ | $\approx 0.2680$ | 1.25 |
| 10 | $\approx 0.4006$ | $\approx 0.0329$ | $\approx 0.0329$ | $\approx 0.0329$ | $\approx 0.0329$ | $\approx 0.0329$ | $\approx 0.0329$ | $\approx 0.4006$ | 2.50 |

Table 2: The distributions for different values of $\epsilon$ by the measurement-based exponential mechanism with $\Delta u = 1$.

| $\epsilon$ | $P_{EM}(0\|\rho_{\vec{v}})$ | $P_{EM}(1\|\rho_{\vec{v}})$ | $P_{EM}(2\|\rho_{\vec{v}})$ | $P_{EM}(3\|\rho_{\vec{v}})$ | $P_{EM}(4\|\rho_{\vec{v}})$ | $P_{EM}(5\|\rho_{\vec{v}})$ | $P_{EM}(6\|\rho_{\vec{v}})$ | $P_{EM}(7\|\rho_{\vec{v}})$ | $\epsilon_D$ |
|---|---|---|---|---|---|---|---|---|---|
| $\infty$ | 0.5 | 0 | 0 | 0 | 0 | 0 | 0 | 0.5 | $\infty$ |
| 1 | $\approx 0.1770$ | $\approx 0.1076$ | $\approx 0.1076$ | $\approx 0.1076$ | $\approx 0.1076$ | $\approx 0.1076$ | $\approx 0.1076$ | $\approx 0.1770$ | 0.50 |
| 3 | $\approx 0.2990$ | $\approx 0.0668$ | $\approx 0.0668$ | $\approx 0.0668$ | $\approx 0.0668$ | $\approx 0.0668$ | $\approx 0.0668$ | $\approx 0.2990$ | 1.50 |
| 5 | $\approx 0.4010$ | $\approx 0.0329$ | $\approx 0.0329$ | $\approx 0.0329$ | $\approx 0.0329$ | $\approx 0.0329$ | $\approx 0.0329$ | $\approx 0.4010$ | 2.50 |
| 10 | $\approx 0.4901$ | $\approx 0.0033$ | $\approx 0.0033$ | $\approx 0.0033$ | $\approx 0.0033$ | $\approx 0.0033$ | $\approx 0.0033$ | $\approx 0.4901$ | 5.00 |

Table 3: The distributions for different values of $\epsilon$ by the measurement-based exponential mechanism with $\Delta u = 1/2$.

ity $p$ to alter quantum measurements and ensure privacy. We investigate the correlation between the neighboring parameter $\eta$ and the privacy budget $\epsilon$ illustrated in Fig. 11 for $p \in \{0.2, 0.3, 0.4\}$. The figure shows that an increase in $\eta$ suggests a broader consideration of neighboring quantum states, making it harder to safeguard privacy from these neighboring states (resulting in an increase in $\epsilon$). This observation highlights the compromise between the relationship of $\eta$-neighbors and the DP budget $\epsilon$.

| Circuit Type | Mechanism | 10 Qubits | 12 Qubits |
|---|---|---|---|
| Random Circuits (RC) | MBEM | $\approx 0$s | 1s |
| | Depolarizing noise | 27s | 7min 7s |
| Variational Quantum Circuits (VQC) | MBEM | $\approx 0$s | 3s |
| | Depolarizing noise | 34s | 10min 15s |

Table 4: Execution times for privacy-utility trade-off analysis of MBEM and depolarizing noise mechanisms on RCs and VQCs.

## K  PRIVACY-UTILITY TRADE-OFF FOR DIFFERENT MECHANISMS AND CIRCUITS

In this section, we present privacy-utility trade-offs for both the MBEM and depolarizing noise mechanisms applied to two types of quantum circuits: random circuits and variational quantum circuits (VQC). The experiments were conducted with varied qubit numbers to evaluate the performance of these mechanisms in multi-qubit scenarios. Specifically, we aim to explore how the hybrid differential privacy framework behaves under classical and quantum noise in multi-qubit settings.

### K.1  EXPERIMENT SETUP AND CIRCUIT CONFIGURATIONS

In these experiments, we utilize two types of quantum circuits: **random circuits** and **variational quantum circuits (VQC)**. Their configurations are described as follows:

- **Random Circuit**: The random circuit consists of 3 to 12 qubits, with a depth randomly chosen between 3 and 5 layers. Single-qubit gates (e.g., $H$, $X$, $Z$, $R_x(\theta)$, $R_z(\theta)$) and $CX$ gates are randomly applied, where the rotation angles $\theta$ are uniformly sampled from $[0, 2\pi]$. This circuit serves as a baseline to evaluate the performance of structured circuits like VQC.

- **VQC**: The variational quantum circuit (VQC) is widely used in quantum machine learning as it serves as a quantum analog of classical neural networks, enabling parameter optimization for specific tasks. In our experiments, the VQC consists of 3 to 12 qubits and multiple layers proportional to the number of qubits. Each layer includes parameterized $R_y(\theta)$ rotation

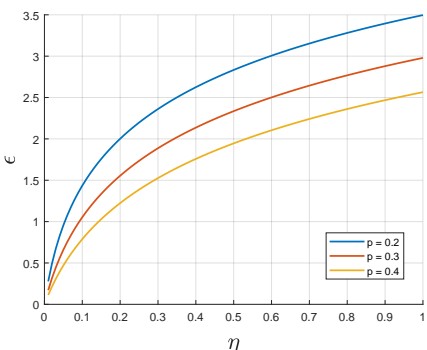

Figure 11: Trade-off between $\epsilon$-DP and $\eta$-neighboring relationship under depolarizing noise with noisy probability $p$.

gates with randomly initialized parameters applied to each qubit, followed by CNOT gates that entangle neighboring qubits.

For both types of circuits, the input states were initialized as classical binary strings, and the output states were simulated using the `Statevector` method to compute exact probability distributions. Errors were introduced into the circuits using the MBEM and depolarizing noise mechanisms, resulting in noisy output distributions. The utility loss was quantified by computing the KL divergence between the original (noise-free) distributions and the noisy output distributions, while the privacy guarantee was assessed by varying the privacy parameter $\epsilon$.

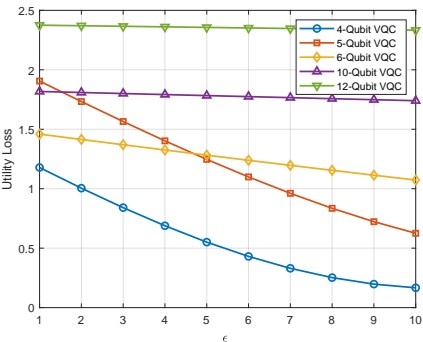

Figure 12: Privacy-utility curves for MBEM on variational quantum circuits (VQC) with varied qubit numbers.

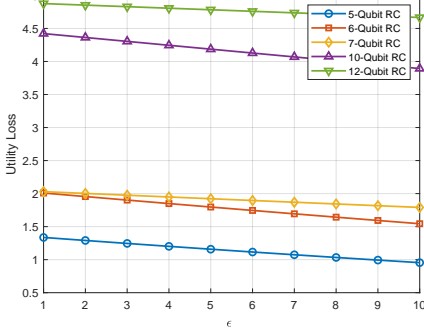

Figure 13: Privacy-utility curves for MBEM on quantum random circuits with varied qubit numbers

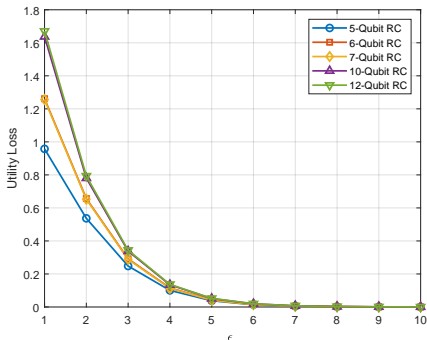

Figure 14: Privacy-utility curves for depolarizing noise mechanism on quantum random circuits with varied qubit numbers

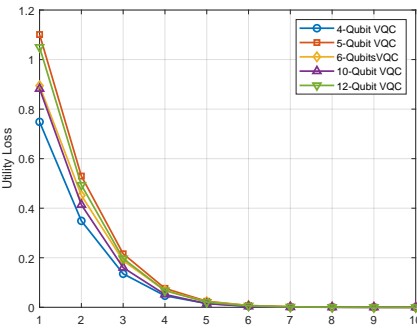

Figure 15: Privacy-utility curves for depolarizing noise mechanism on variational quantum circuits (VQC) with varied qubit numbers

The results of these experiments are shown in Figures 12[1] to 15, which illustrate the privacy-utility trade-offs for the two mechanisms applied to random circuits and VQC, respectively. Specifically:

- **Figures 12 and 13** present the privacy-utility trade-offs for MBEM applied to random circuits and VQC, respectively. The results demonstrate that as privacy protection weakens ($\epsilon$ increases), utility loss decreases, indicating that reduced privacy leads to improved usability. Furthermore, the utility of the algorithm decreases as the number of qubits increases, with the degradation more evident in MBEM compared to the depolarizing noise mechanism.

- **Figures 14 and 15** display the corresponding privacy-utility trade-offs for the depolarizing noise mechanism. Similar to MBEM, weaker privacy protection improves utility. However, the impact of qubit numbers on utility is less significant for the depolarizing mechanism compared to MBEM.

### K.2 ANALYSIS AND CONCLUSIONS

From these results, we observe that the depolarizing noise mechanism exhibits a privacy-utility trade-off that is less sensitive to the number of qubits, making it more predictable in multi-qubit scenarios. On the other hand, MBEM is more sensitive to qubit numbers, indicating that it requires detailed privacy-utility trade-off analyses tailored to specific qubit configurations when applied in practice.

Table 4 compares the execution times of the two mechanisms for different qubit numbers. MBEM demonstrates significantly faster execution, completing analyses within seconds, even for 12 qubits. In contrast, the depolarizing mechanism requires much longer computational times as the qubit number increases. For example:

---

[1]Fig.12 is a copy of Fig. 9.

- On random circuits with 10 qubits, MBEM completes in almost 0 seconds, whereas the depolarizing noise mechanism requires 27 seconds. For 12 qubits, MBEM takes 1 second, but the depolarizing noise mechanism requires 7 minutes and 7 seconds.

- On VQC circuits with 10 qubits, MBEM completes in almost 0 seconds, while the depolarizing noise mechanism requires 34 seconds. For 12 qubits, MBEM takes 3 seconds, whereas the depolarizing noise mechanism requires 10 minutes and 15 seconds.

In summary, these findings highlight two key takeaways:

1. The depolarizing mechanism's utility-privacy trade-off is less affected by the qubit number of quantum systems, making it a robust choice in multi-qubit scenarios.

2. MBEM, while more sensitive to qubit number, offers extremely fast execution times, allowing for efficient privacy-utility trade-off analyses without significantly impacting performance. This makes MBEM suitable for balancing privacy and utility in time-sensitive applications where computational efficiency is critical.

