# OpenReview forum: "Differential Privacy of Hybrid Quantum-Classical Algorithms"
_ICLR.cc/2026/Conference — Submitted to ICLR 2026_

### Official Review · Reviewer_emrU · 2025-10-30

**Soundness:** 3
**Presentation:** 4
**Contribution:** 2
**Rating:** 4
**Confidence:** 3

**Summary:**

This paper considers the problem of making quantum computations differentially private. There are three kinds of privacy that they relate:
* The outcome of a classical randomized function is DP
* The resulting quantum state after processing through a quantum channel is DP
* The outcome of a quantum measurement is DP

The paper shows the natural analog of the standard post-processing theorem and composition theorem: Applying a randomized function to the output of a quantum DP measurement is DP. Combining two measurements preserves DP, with additive error and failure probability.

The paper considers how to make quantum measurements DP. They consider 1) an exponential mechanism applied to a *classical* representation of the distribution of the quantum measurement. 2) Naturally occurring noise in quantum computers. They show what level of noisy probability is needed to achieve various parameters of DP. They also compare these on several experiments.

**Strengths:**

* I don't work in this area, but (assuming their related work is comprehensive) they are the first to consider a DP definition for quantum measurements. They also nicely relate this notion of DP to other notions of DP, and extend the standard post-processing and composition theorems mentioned above.

* Of course, the main issue of physical quantum computers is the noise. So it's nice to see the silver lining of it as a DP mechanism.

**Weaknesses:**

* All of their "theorems" (except maybe theorem 6, which just seems like a bunch of definitions and algebra) seem completely obvious and straightforward to prove. I think there's limited technical contribution here.

* The exponential mechanism they propose seems to require classical access to the distribution over quantum measurements (e.g., they say they can empirically approximate it if it's infeasible to represent analytically). This seems less than ideal because, if we knew the distribution over quantum measurements, we wouldn't need the quantum computer and we could just sample from the distribution classically. Hence, their proposed mechanism can't be used on the problems where we'd actually need a quantum computer to solve it.

Minor:

* Figure 1 and Figure 3 don't seem to have rendered correctly; there are oddly shaped boxes with nothing in them.

**Questions:**

Is there any result in the paper that is technically challenging to prove? Or requires new insights?

Am I correct that your exponential mechanism requires classical access to the quantum measurement?

Overall: I don't work this area, so I'll defer to other reviewers on the value of their DP framework and the connections they describe. If the other reviewers disagree with me about the technical contribution in this work, or the value of its framework, I would be willing to raise my score.

---

### Official Review · Reviewer_Rvso · 2025-10-31

**Soundness:** 2
**Presentation:** 1
**Contribution:** 2
**Rating:** 4
**Confidence:** 2

**Summary:**

The paper studies differential privacy (DP) in hybrid quantum–classical algorithms. The key insight is to introduce DP for a fixed quantum measurement and connect it to DP for the quantum circuit, with composition handled via neighbor-preserving quantum encodings. The authors then propose two practical mechanisms to enforce privacy at the measurement stage: (i) adding quantum depolarizing noise before measurement, and (ii) a measurement-based exponential mechanism (MBEM) as a classical post-processing step. Using neighboring-preserving encodings and a Heisenberg-picture analysis, the work links this measurement-level DP to both classical DP and quantum DP. The authors also run small-scale experiments (GHZ states, random circuits, and VQCs up to ~12 qubits) to illustrate privacy–utility trade-offs, and they suggest that for a fixed measurement the depolarizing mechanism can yield tighter guarantees than standard quantum differential privacy.

**Strengths:**

- The paper specifically addresses privacy in hybrid quantum–classical computation, representing a clear element of novelty.


- The authors propose methods that provably achieve a chosen level of differential privacy, with tunable parameters to reach arbitrary targets.


- Numerical experiments are provided in support of the theory, illustrating the privacy–utility trade-offs.

**Weaknesses:**

Although the above strengths hold, I think the current version does not meet the ICLR bar for the following reasons:

- The presentation is not very clear. The definitions of classical and quantum differential privacy are stated only in technical terms, with little intuition for what they mean in practice. In addition, there is no precise definition of neighbor-preserving states/encodings.


- There are no proof sketches or intuitive explanations of why the proofs work, nor a discussion of the implications of the main theorems.


- The advantage of the proposed measurement-level DP over QDP is shown only empirically (e.g., in Fig. 6). There is no corresponding theoretical claim or deeper analysis to delineate when and why it should hold.


- There is no experiment on a complete learning task; the paper lacks an end-to-end demonstration that integrates the privacy mechanisms into a full training/evaluation pipeline.

**Questions:**

- Could the authors better introduce the objective of differential privacy mechanisms and briefly comment on the key definitions introduced in the paper?


- Could the authors explain the implications of their theorems in accessible language and add a proof sketch/intuition for why the results hold?


- Could the authors include an end-to-end learning experiment that involves a full training loop using their privacy mechanisms?

---

### Official Review · Reviewer_UV5E · 2025-11-01

**Soundness:** 3
**Presentation:** 3
**Contribution:** 3
**Rating:** 4
**Confidence:** 3

**Summary:**

This paper proposes a new theoretical framework for ensuring the privacy of hybrid quantum-classical algorithms. We newly define the concept of differential privacy in quantum measurements, which focuses on the quantum measurement stage, the key point at which privacy leakage occurs.

And this paper proves that this technique, consisting of (i) quantum depolarization noise, and (ii) measurement-based exponential mechanisms, provides more efficient and robust privacy guarantees than conventional general quantum DP.

**Strengths:**

This paper is a very fundamental and original contribution that resets the scope of QML privacy discussions to address the privacy challenges of hybrid QML.

This paper shows that the amount of noise required for privacy guarantees in hybrid algorithms can be substantially reduced, greatly improving the usefulness of the model.

Through Theorme, theoretical completion is increased, and the performance of the technology that the authors want to explain is highly communicated through the experimental results of the figures and tables.

**Weaknesses:**

There are some shortcomings in terms of the practicality and efficiency of the core ideas of the paper.

The cost of calculating the privacy budget of DP of QM (Theorem 6) with quantum noise increases exponentially with the number of qubits.
It is interpreted as a limitation that the proposed technology is only valid on small systems. It is necessary to explain how the proposed technology can be applied and how it works when the system becomes large.

The practicality of MBEM is unclear due to the lack of discussion on how to accurately and efficiently calculate the probability distribution difference between two neighboring states ($\rho \sim \sigma$) in complex VQC or Hamiltonian measurements.

The authors are abstractly defining neighboring quantum states ($\rho \sim \sigma$). This paper requires analysis and guidelines on how they define neighbours and how they affect the techniques proposed by definition.

**Questions:**

The cost of analyzing the privacy budget of the proposed mechanism itself seems inefficient. Is there a limit that cannot be used in large systems? Please explain how this mechanism can be applied to practical hybrid algorithms.


How was the value of the sensitivity $\Delta u$ used in the experiment derived?


How does ordinary angle encoding transform the neighborhood relations of classical data into a tracking distance $\eta$-neighbor relationship of quantum states?

---

### Official Review · Reviewer_Jw6L · 2025-11-01

**Soundness:** 2
**Presentation:** 3
**Contribution:** 2
**Rating:** 4
**Confidence:** 4

**Summary:**

This paper introduces a novel framework, "Differential Privacy of Quantum Measurements" (DP of QM), to address the privacy gap in hybrid quantum-classical (HQC) algorithms relevant to the NISQ era. The framework focuses on the measurement step as the critical quantum-classical interface. The authors provide a formal (ϵ,δ)-DP definition for quantum measurements and establish corresponding post-processing and composition theorems. Two mechanisms are proposed: (1) a pre-measurement quantum depolarizing noise, with bounds (Theorem 6) dependent on the specific measurement's spectral properties, and (2) a post-measurement classical "Measurement-Based Exponential Mechanism" (MBEM) that uses outcome probabilities as utility scores. Numerical results show this framework yields tighter bounds than general QDP and that MBEM is computationally highly efficient.

**Strengths:**

1.	The paper correctly identifies a practical and important gap. Privacy for hybrid algorithms is far more relevant in the NISQ era than pure QDP, and focusing on the measurement interface is the correct approach.
2.	The proposed Measurement-Based Exponential Mechanism (MBEM) is a key strength. It is a clever and elegant adaptation of the classical exponential mechanism to the quantum measurement context. Using the measurement probabilities directly as utility scores is a natural fit for discrete outcomes and avoids the issues of applying continuous (e.g., Gaussian) noise.
3.	The empirical results clearly show that MBEM is extremely computationally efficient, taking only seconds to run on 12-qubit simulations. This contrasts sharply with the "quantum" depolarizing noise approach, which is shown to be computationally slow. This makes MBEM a genuinely practical proposal.
4.	The paper makes an important conceptual point: QDP must be private against all possible measurements, whereas HQC algorithms use a fixed, known measurement. The authors rightly show (Theorem 6, Fig. 6) that by focusing on the specific measurement, one can derive much tighter (i.e., better) privacy bounds than the general QDP framework allows.

**Weaknesses:**

1.	The paper's most significant weakness is the poor scalability of its best mechanism, MBEM. The empirical results in Figures 9 and 12 clearly show that the utility loss (KL divergence) of MBEM increases dramatically with the number of qubits. For 12 qubits, the utility loss is already very high (KL > 1.5). This suggests that for any practically-sized quantum processor (e.g., 50+ qubits), the utility loss would be total, making the mechanism unusable.
2.	The depolarizing noise mechanism, while presented as the quantum alternative, appears completely impractical. Its DP bounds (Theorem 6) and computational analysis (Appendix J, Table 4) depend on the spectral properties (λ_max,λ_min) of the POVM operators. Calculating these is computationally hard and scales exponentially. The paper's own data shows this takes 10+ minutes for just 12 qubits. This approach is not scalable.
3.	The utility of MBEM is shown to be highly dependent on finding a tight sensitivity bound Δu. The paper uses Δu=1 as a loose default and Δu =1/2 as a "refined" value for its example. However, it provides no general algorithm or procedure for a user to find the tightest possible Δu for an arbitrary new HQC measurement. This is a critical missing methodological component.
4.	The quantum noise analysis is only for the depolarizing channel. This is a common simplification but is not representative of the complex, structured noise in real NISQ devices. This limits the applicability of the quantum-related claims (e.g., Theorem 6).

**Questions:**

1.	The data in Figure 9/12 shows that the utility loss of MBEM scales very poorly with the number of qubits. Is this not a fatal flaw for the mechanism? Does this imply MBEM is fundamentally unsuitable for any HQC algorithm of practical (e.g., >20 qubit) scale?
2.	The analysis in Appendix J.1 shows that the utility of MBEM is critically dependent on finding a tight Δu. You use Δu=1/2 for an example, but what is the general procedure to calculate the tightest Δu for an arbitrary, complex VQC measurement? Without such a procedure, how can a user practically apply your mechanism?
3.	The computation for the depolarizing noise mechanism (Theorem 6) requires spectral analysis, which your own data (Table 4) shows is exponentially slow (10+ minutes for 12 qubits). Given this, how can this be considered a practical or scalable mechanism? Isn't it computationally less practical than MBEM?
4.	The initially dismiss Laplace/Gaussian noise as "unsuitable" for discrete outcomes, but then you compare against them in Figure 8 and Appendix I. Could you clarify how they were adapted? Is the superior performance of MBEM fundamental, or is it an artifact of the chosen utility metric (KL Divergence)?

---

### Meta-Review · Area_Chair_pAqG · 2026-01-10

**Summary:**

The reviewers raised concerns about the scalability and practicality of the proposed framework. A primary issue is the poor scaling of the Measurement-Based Exponential Mechanism (MBEM), where utility loss becomes prohibitive as the qubit count increases, combined with the exponential computational cost required to determine privacy parameters for the depolarizing noise mechanism. Moreover, reviewers questioned the theoretical depth and novelty, noting that some theorems appear straightforward and that key concepts such as neighboring quantum states and sensitivity bounds lack clear definitions or practical guidelines for calculation. The absence of end-to-end learning experiments and the reliance on simplified noise models also limits the demonstration of the framework's real-world applicability. In its present form, the paper does not meet the bar for publication at ICLR.

**Reviewer Concerns:**

Not applicable since no rebuttal was submitted by the authors.

**Reviewer Scores:**

Not applicable since no rebuttal was submitted by the authors.

---

### Decision · Program_Chairs · 2026-01-26

Reject